# In-Silico and In-Vitro Analysis of the Novel Hybrid Comprehensive Stage II Operation for Single Ventricle Circulation

**DOI:** 10.3390/bioengineering10020135

**Published:** 2023-01-19

**Authors:** Arka Das, Marwan Hameed, Ray Prather, Michael Farias, Eduardo Divo, Alain Kassab, David Nykanen, William DeCampli

**Affiliations:** 1Department of Mechanical Engineering, Embry-Riddle Aeronautical University, Daytona Beach, FL 32114, USA; 2Department of Mechanical Engineering, American University of Bahrain, Riffa 942, Bahrain; 3Department of Mechanical and Aerospace Engineering, University of Central Florida, Orlando, FL 32816, USA; 4The Heart Center at Orlando Health Arnold Palmer Hospital for Children, Orlando, FL 32806, USA; 5Department of Clinical Sciences, College of Medicine, University of Central Florida, Orlando, FL 32816, USA

**Keywords:** congenital heart defect, hypoplastic left heart syndrome, hybrid comprehensive stage II, in-silico modeling, in-vitro modeling, multiscale modeling, multi-object tracking, kalman filter

## Abstract

Single ventricle (SV) anomalies account for one-fourth of all congenital heart disease cases. The existing palliative treatment for this anomaly achieves a survival rate of only 50%. To reduce the trauma associated with surgical management, the hybrid comprehensive stage II (HCSII) operation was designed as an alternative for a select subset of SV patients with the adequate antegrade aortic flow. This study aims to provide better insight into the hemodynamics of HCSII patients utilizing a multiscale Computational Fluid Dynamics (CFD) model and a mock flow loop (MFL). Both 3D-0D loosely coupled CFD and MFL models have been tuned to match baseline hemodynamic parameters obtained from patient-specific catheterization data. The hemodynamic findings from clinical data closely match the in-vitro and in-silico measurements and show a strong correlation (r = 0.9). The geometrical modification applied to the models had little effect on the oxygen delivery. Similarly, the particle residence time study reveals that particles injected in the main pulmonary artery (MPA) have successfully ejected within one cardiac cycle, and no pathological flows were observed.

## 1. Introduction

Single ventricle is one of the complex forms of cyanotic congenital heart defect (CHD). Studies report that neonates with single functioning ventricle have a significantly high mortality rate during the early years of life [1]. Patients born with variants of SV CHD require immediate surgical treatment after birth. This initial stage operation is followed by two additional staged surgical procedures to sustain life [2,3,4,5]. Neonates with some SV variants have a hypoplastic systemic outflow tract, aortic annulus, and aortic arch, each of which impedes systemic blood flow [4]. These anatomic malformations result in inherently unstable physiology, in which the systemic circulation is dependent on the patency of the ductus arteriosus whilst the pulmonary blood flow is “unguarded” and tends to become excessive. The objective of the first stage of treatment is to establish an adequate and relatively balanced systemic and pulmonary circulations [6]. In the subsequent staged operations, the therapeutical goal is to establish a series circulation whereby the remaining ventricle powers the systemic blood flow, and the pulmonary circulation becomes passively driven by the pressure gradient between venous return and the atrium.

Neonates undergo the Stage I procedure, the Norwood, immediately after birth. During this stage, the anomalous systemic circulation is reconstructed to achieve unobstructed and equal flow (on average) to systemic and pulmonary circulation. In this stage, atrial septectomy is performed, and the diminutive aorta is reconstructed and connected to the single ventricle at the pulmonary root. A Blalock Taussig Thomas (BTT) shunt is implanted between the innominate artery and the pulmonary arteries to create a parallel path between the systemic and the pulmonary circulation [7,8]. Stage II surgery, the bidirectional Glenn (BDG), is performed between three and six months. In this procedure, the superior vena cava (SVC) is disconnected from the right atrium (RA) and connected directly to the pulmonary arteries to supply pulmonary flow passively [9]. The main goals of this operation are to provide more stable pulmonary blood flow by a vessel that can grow and relieve the excess volume load on the single functioning ventricle. In Stage III surgery, the Fontan procedure [10], the inferior vena cava (IVC) is decoupled from the RA, and the venous flow is routed to the pulmonary arteries typically through a synthetic conduit (“Fontan conduit”) typically placed outside the heart.

Following this operation, a series circulation (systemic to pulmonary circulation) driven by the right ventricle (RV) is achieved. Despite a significant improvement in survival rate after the palliative procedures, an early mortality rate and poor functional outcomes in the survivors are significantly high [9]. Due to the increased mortality, morbidity, and surgical trauma associated with the conventional stage I procedure [11,12], the hybrid Norwood procedure was devised as an alternative technique for performing the stage I procedure in neonates with HLHS [13,14,15]. This approach avoids the prolonged use of cardiopulmonary bypass (CPB), deep hypothermia, and circulatory arrest, which are often associated with high morbidity [4,16,17]. The hybrid Norwood procedure establishes a relatively stable circulation by the stenting of the ductus arteriosus and banding (intentional narrowing) of the left pulmonary artery (LPA) and right pulmonary artery (RPA) branches. A balloon atrial septostomy is also performed as required. Although the purported advantage of the hybrid technique is to minimize the extent of surgical trauma in neonates [17,18,19] by avoiding CPB, hypothermia, and circulatory arrest, these techniques are typically needed while performing the subsequent “comprehensive stage II” procedure [13,14,15,20]. In this operation, reconstruction of the aortic arch and pulmonary artery and management of a ductal stent are required in addition to creating the BDG [17,18]. In many centers, the mortality of this operation is significantly higher than that of the standard BDG and, in some cases, comparable to the standard Norwood procedure.

Reducing the complexity of the second stage operation following the hybrid Norwood stage I would reduce morbidity and mortality and sustain the putative advantages of the hybrid approach. DeCampli et al. [21] have described a novel alternative hybrid technique to the conventional comprehensive stage II, the “hybrid comprehensive stage II” procedure, to reduce surgical trauma (Figure 1). The HCSII technique applies to the subset of HLHS patients with sufficient native antegrade aortic flow for coronary and upper systemic perfusion. This novel procedure avoids the reconstruction of the aortic arch and the creation of the Damus-Kaye-Stansel (DKS) connection. The antegrade flow originates from a ventricular septal defect (VSD), allowing blood to perfuse the upper circulation (Figure 1). This procedure involves placing a stented baffle within the main pulmonary artery joining the left branch ostium to the right branch ostium to separate systemic and pulmonary flow (Figure 1) [21,22,23]. Also, the operation typically entails re-stenting of the ductus to maintain unobstructed lower body systemic circulation. Finally, a bidirectional superior cavopulmonary connection (analogous to the BDG) completes the HCSII procedure. The procedure can be done without deep hypothermia or selective cerebral perfusion and is completed in a significantly shorter time than the usual comprehensive stage II operation. A follow-up review of patients undergoing this approach was recently published by Farias et al. [22].

As direct in-vivo measurements are often difficult to obtain, the understanding of the outcomes of the proposed surgical procedure and the impact of reconstructed patient-specific anatomy on the crucial clinical, e.g., systemic and pulmonary hemodynamics, thrombosis, particle residence time, and power loss and wall shear stress, etc. must be studied using in vitro and/or in silico techniques. The multiscale modeling framework involving computational and experimental simulations can closely mimic the circulation in the reconstructed anatomy and extract the local and global hemodynamic parameters to cross-validate the vital observations. In recent years, in-silico and in-vitro modeling techniques have been widely used to qualify and quantify the hemodynamics of surgical interventions and used as a treatment-planning tool for SV anomalies [24,25,26,27,28,29,30,31,32,33,34].

Hameed et al. [23] investigated the pressure drop across the HCSII baffle and the potential for vortex formation using a multiscale CFD model. The study showed that pressure drops across the baffle were within the clinically acceptable values for a range of baffle cross-section narrowing. The current study focused on the potential effects of various levels of baffle narrowing and ascending aorta diameter ranges on oxygen transport, power loss, and particle residence time in the novel HCSII circulation. The in-silico models of the synthetic HCSII representative geometries with various baffle and ascending aorta dimensions are used to quantify the resultant hemodynamics. The in vitro study consists of a dynamically scaled MFL integrating a representative synthetic phantom of the reconstructed HCSII anatomy. The model incorporates an intra-pulmonary baffle graft and a Palmaz Genesis (PG) 2510B stent (Cordis, Bridgewater, NJ, USA). In-vitro experiments cross-validate the key in-silico findings while matching the catheter reports provided by our clinical collaborators.

## 2. Materials and Methods

It is crucial to implement a multiscale 0D-3D CFD coupling scheme that allows the investigation of surgical planning in a virtual physics-based environment. A multiscale CFD model was developed to investigate the effect of a range of surgically viable geometrical alterations of the HCSII on the hemodynamics of this palliative procedure. The multiscale model couples a 3D CFD model of the synthetic HCSII anatomy with a 0D lumped parameter model (LPM) of the peripheral circulation. To cross-validate the CFD findings, it is essential to have a robust benchtop setup that can emulate the physiologically consistent flow field. Hence, a dynamically scaled MFL setup that integrates a representative 3D printed replica of the reconstructed HCSII anatomy incorporating an intra-pulmonary baffle graft and a PG 2510B stent has been developed. To preserve consistency and compare the in-silico and in-vitro findings, the experimental study followed the same protocol as the computational study.

### 2.1. Anatomical Model

#### 2.1.1. In-Silico Model

The 3-dimensional baseline synthetic HCSII geometry was modeled using SolidWorks (Dassault Systemes, Concord, MA, USA), which includes the MPA, ascending aorta (AA), the descending aorta (DA), right and left subclavian arteries (RSA, LSA), right and left carotid arteries (RCA, LCA), right and left coronary arteries (RcorA, LcorA), and the BDG procedure with SVC connected to the RPA and LPA as shown in Figure 2. The in-silico model was developed based on the averaged dimensions obtained from the deidentified angiographic images of three patients provided by our clinical collaborators at Arnold Palmer Hospital (APH) for children.

To study the effect of the MPA narrowing on systemic saturations, power loss, and particle residence time, the nominal geometry was manipulated to generate (1) four MPA narrowing variations (Figure 3a), (2) four ascending aorta root diameter $\left(D_{AA}\right)$ variations (Figure 3b). Table 1 summarizes each model variant.

##### Mesh

The geometries were imported into the destination CFD software STAR-CCM+ (Siemens, Munich, German). A high-quality finite volume mesh was generated following a grid independence study using tetrahedral elements [35], with a base size of 0.3 mm and three prism layers at the wall boundary (Figure 4). The nominal HCSII geometry was modeled with 100% stenosis to replicate the complete obstruction of the distal arch. The number of cells varied between 1.5 million and 2.3 million across all geometries modeled.

An extensive grid independence analysis was carried out to calculate the optimum mesh density of the in-silico models, as described in our previous studies [23,29]. This analysis was performed by running steady-state simulations and monitoring the pressure and the mass flow rate at the different locations (MPA root, SVC, RPA, DA, AO, and LPA) of the CFD domain. In this study, three different levels of mesh refinements were carried out, and relative percentage changes were evaluated at those respective locations in the CFD domain, which were shown to be lesser than 0.15% between the current mesh and the next coarsest mesh, as shown in [29] (pp. 49–50).

#### 2.1.2. In-Vitro Phantom

The 3D HCSII phantom was designed using CATIAV5 (Dassault Systèmes) for conducting in-vitro simulations. The synthetic HCSII phantom was modeled using the deidentified angiographic images of a patient with 0.34 m^2^ body surface area (BSA) who underwent the HCSII procedure. The average dimensions of each vessel were derived from the respective angiographic images and used to construct a synthetic 3-D centerpiece representing the HCSII circulation, as shown in Figure 5A–F and Appendix A. The experimental 3-D phantom was developed in a two-part process; the top half of the phantom contained a major section of the conduits derived from the angiogram data. Dimensions used for constructing the experimental model are provided in Appendix A. The 3-D phantom (Figure 5C and Appendix A) includes three distinct regions, (1) Upper circulation across the native underdeveloped aorta, (2) lower body circulation across the MPA (neo-aorta) to the descending aorta, and (3) the venous return to the lungs. To simplify the upper body peripheral circulation, the vascular beds of RSA, RCA, LSA, and LCA conduits are lumped together. The diameter of the resultant conduit was the summation of RSA, RCA, LSA, and LCA diameters (Figure 5A).

In keeping with the HCSII surgical procedure, the phantom (1) does not include the patent ductus arteriosus (100% stenosis, Figure 6E), (2) no DKS connection, and (3) retains the native aortic arch [21,22,23]. This centerpiece incorporates the insertion of a single PG 2510B stent placed in the pulmonary artery (PA) conduit and covered with a baffle made of expanded polytetrafluoroethylene (ePTFE) graft material which was partially sutured within the MPA conduit to keep systemic and pulmonary flows separated. The designed 3-D centerpiece integrated with the MFL preserved the characteristic features of the reconstructed anatomy.

A hollow elliptical section was designed on the MPA conduit of the centerpiece, as shown in Figure 6C,F to place the baffle in the MPA conduit that closely replicated the stented baffle configuration in the reconstructed HCSII anatomy mentioned in [20,21,22]. The sutured end of the baffle on the MPA conduit constrained the stent migration in the PA conduit. Barbed conduits have been designed at six locations to integrate the centerpiece with the MFL. The pressure sensors were placed at three key locations, i.e., MPA, AA, and DA conduits in the centerpiece to acquire the hemodynamic pressure parameter for each cardiac cycle during in-vitro simulations (Figure 6A,B). The bottom half of the centerpiece, as shown in Figure 6C, covered the PA section once the baffle and the stent were placed in their respective locations. In Figure 6D, Section 1 forms the MPA, Section 2 forms the AA conduit, and Section 4 represent the flow towards the upper body. Section 3 and Section 5 are connected together to form the PA. Section 6 forms the DA in the 3-D in-vitro phantom.

### 2.2. Lumped Parameter Model

The lumped parameter model has been widely used to provide a simplified representation of the cardiovascular system and to quantify hemodynamics variables, i.e., pressure and flow rates in the regions of interest [36,37,38,39]. The LPM is an electric circuit analogy used to model the peripheral circulation of the human circulatory system. Using this analogy, blood flow through various vessels in the circulatory system can be lumped and represented as electrical circuits that include resistors (R), capacitors (C), and inductors (L) [29,30,31,32,40]. These elements represent vascular resistance, vascular compliance, and vascular inertance, respectively. LPM is mainly used to replicate and couple the dynamics of the peripheral circulation (0D system) with the flow field generated within the 3-D model.

#### 2.2.1. LPM Setup for In-Silico Model

A complex circulatory system can be modeled using an electric analog circuit known as LPM. The LPM was tailored to replicate the peripheral circulation of the patient’s anatomy and tuned to closely match the catheter data. The HCSII circulation was compartmentalized in a series of arterial and venous elements that include resistors, capacitors, and inductors as discussed in [23,29]. The LPM-generated waveforms used as boundary conditions (BCs) to drive the CFD model. The model consists of three main subsystems: the upper circulation, the lower circulation, and the pulmonary circulation, as shown in Figure 7. The RV powers the LPM circuit utilizing a time-varying capacitor regulated by the elastance function $E_{n}(t_{n})$ [41,42], which is defined as
(1)$$E_{n}\left(t_{n}\right)=\left[\frac{{\left(\frac{t_{n}}{0.303}\right)}^{1.32}}{1+{\left(\frac{t_{n}}{0.303}\right)}^{1.32}}\right]\left[\frac{1}{1+{\left(\frac{t_{n}}{0.508}\right)}^{121.9}}\right]$$

The valves are represented as diodes in the LPM, mathematically modeled as Heaviside step functions responding to the pressure gradient across the valve.

The pressure drop across a segment is given by
(2)$$\Delta P=l\frac{dQ}{dt}+RQ$$
and the flow rate across compliance is
(3)$$Q=C\frac{dP}{dt}$$
where *Q* is the flow rate, and Δ*P* is the pressure difference. These differential equations are derived using the hydraulic analogies shown in Figure 7, along with the conservation of current (KCL) and voltage laws (KVL). Kirchhoff laws state that the current entering a junction or a node is equal to the current exit it, and the sum of voltages in a closed-loop has to be zero, as described in the equations in Appendix B. From the LPM, 34 coupled ordinary differential equations were derived and solved using an in-house fourth order adaptive Runge-Kutta solver. The VSD was only modeled in the LPM circuit as a non-linear resistance in the circuit between the inlet of MPA and the inlet to AA.

The LPM was then tuned to obtain the desired waveforms matching catheter data [37,43,44]. The target cardiac output (CO) was 2.99 L/min, with 60% perfusing to the lower body and 40% perfusing the upper body. Moreover, the model implements the coronary flow with a 70–30% split between the left and right coronary arteries to retain physiological accuracy [45]. In this study, each face of the in-silico geometry in the CFD domain was defined with a specific type of boundary condition (Stagnation pressure inlet, mass flow inlet, and outlet). Table 2 details the various types of BCs imposed for each inlet and outlet in the CFD domain.

Figure 8 shows a sample set of BC waveforms that were applied to the CFD model following the LPM tuning. Each outlet was treated as a mass flow outlet, while the aortic root was considered as a mass flow inlet, and a total pressure was imposed at the MPA.

#### 2.2.2. LPM Setup for In-Vitro Model

The reduced LPM used in this in-vitro study was derived from the full-scale LPM model of Hameed et al. [23,29]. The full-scale LPM is not practically feasible in a laboratory benchtop realization. Tuning such an MFL with a full-scale LPM circuit would be difficult. Furthermore, the number of elements requiring connectors and tubing would create too much inertance. The full-scale LPM was reduced to a branched LPM of equivalent impedance to create a practically feasible system. An in-vitro study on HCSII circulation was conducted through a dynamically scaled MFL using patient-specific geometry.

The MFL was modeled to resemble the physiological anatomy of a patient with 0.34 m^2^ BSA. In this MFL setup, the lumped compartments representing the peripheral circulations of the patient were connected to the 3-D phantom of the HCSII geometry. The MFL consists of four compartments, and each compartment was developed with two elements (i.e., resistance and compliance) Windkessel models. Four compartments represent upper systemic, lower systemic, right, and left pulmonary circulations, as shown in Figure 9. Each lump represents the arterial and venous vasculature found within the physiological anatomy. The voltage source represents the RV. The detailed design and fabrication procedure for the MFL setup have been detailed in our previous studies [31,32].

The VSD was not modeled in the 3-D phantom but is represented in the LPM circuit as two resistor valves placed between the bifurcation junction (upstream of the RV) and the inlet of the MPA and AA in the 3-D phantom. This study mainly focused on investigating the flow field developed in the reconstructed HCSII geometry; hence, the BDG section was replicated in the LPM circuit, as shown in Figure 9. Every branch in the reduced LPM contains circuit elements that correspond with a device in the MFL that serves as a physical realization of that parameter. The pressure drop across each branch in the MFL was tuned by a needle resistor valve. Vascular compliance in each compartment was achieved by implementing an annular design-based compliance chamber. Table 3 represents the compliance values considered in each lump while conducting the experimental study.

### 2.3. CFD-LPM Coupling

To converge the flow field, a loose coupling mechanism has been implemented between 0D LPM and 3D-CFD domain [46,47,48] (Figure 10). The algorithm regulating the coupling operates at the cardiac cycle level. The LPM generates a set of time-dependent BCs (waveforms), which were imposed on the inlets and outlets of the CFD domains, as given in Table 2. Once the CFD simulation completes three heart cycles, the pressure and flow rates are sampled across the domain at locations corresponding to nodes (pressure) and segments (flow rate) in the LPM. The sample quantities are then time-averaged over all cycles, and the CFD domain resistances are calculated using Ohm’s Law. Once computed, these resistances are updated in the LPM (Figure 10), and the new BCs for the CFD are evaluated. This process is repeated iteratively to converge the flowfield solution between the LPM and the CFD domain, comparing the changes between the pressure waveform at the outlets of the CFD and the LPM. The convergence criteria are set so that the change in the flow in all branches is less than 10^−3^ or resistor values no longer change when iterated.

An automated algorithm that regulates data exchange and storage (Java Macro) carried out this process, which usually takes about 15–20 iterations to achieve convergence. The convergence criteria for the coupling mechanism require an iterative convergence of less than 10^−3^. Following convergence, the CFD computation ran for three more cardiac cycles to gather data for postprocessing.

### 2.4. Fluid Domain Solver

We employed the segregated flow solver in STAR-CCM+ to resolve the flow field using a finite volume solution for Navier-Stokes and continuity equations. The flow was modeled as incompressible (ρ = 1060 kg/m^3^), unsteady, and the fluid was assumed to be non-Newtonian.
(4)$$\nabla \cdot \vec{V}=0$$
(5)$$\rho \frac{\partial \vec{V}}{\partial t}+\rho \left(\vec{V}\cdot \nabla \right)\vec{V}=-\nabla p+\nabla \cdot \sigma$$
(6)$$\sigma =\mu \left(\dot{\gamma }\right)\left[\nabla \vec{V}+{\left(\nabla \vec{V}\right)}^{T}\right]$$

Here, $\vec{V}$ is the velocity, p is the pressure, and Equation (6) represents the viscous stress tensor with a given relation for the viscosity $\mu \left(\dot{\gamma }\right)$ as a function of shear rate $\dot{\gamma }$. A simplified three parameter Carreau-Yasuda model (Figure A2) was adopted to model blood as mentioned by Hameed et al. in [23]. The parameters were determined by curve fitting clinical data assuming a 40% hematocrit [49]. Parameters have been summarized in Table 4.

The time-step used in this simulation is 0.004 s, calculated based on the Courant number $\left(Co\right)$ for an implicit scheme (Co≅1).
(7)Co=uΔt/Δxwhere u is the velocity, Δt is the time-step, and Δx is the mesh base size. This achieves time-accurate results on a grid that has been obtained after a grid convergence study.

The fluid domain in Figure 11 depicts a 100% stenosis as a result of the complete obstruction of the distal arch. For simplicity, the fluid domain has been split into three separate regions, (a) the systemic flow originating from the pulmonary root to the lower body, (b) the systemic flow originating in the aortic root to the upper body, and (c) the SVC flow to the PA as shown in Figure 11.

### 2.5. Oxygen Transport Model

In the HCSII model, flow pumped by the RV is distributed to the lower systemic (S_.L._) circulation and the upper systemic (S_.U._) circulation. The venous blood from the lower circulation flows back through the IVC to the RA, while the upper body deoxygenated blood flows back to the PA through the SVC, leading to two parallel flows. The schematic in Figure 12 represents a 1D oxygen transport model for a generalized HCSII circulation. A similar oxygen transport model has been derived by Santamore et al. [50].

The analysis assumes a steady-state condition. As per the law of mass conservation, systemic oxygen consumption ($C{\dot{V}}_{O2}$) and oxygen uptake ($S{\dot{V}}_{O2}$) must be equal at the cellular level. To distinguish upper and lower body oxygen consumption the competition split ratio parameter x is introduced. Upper body oxygen consumption then can be expressed as $x\;C{\dot{V}}_{O2}$ while lower body oxygen consumption is $\left(1-x\right)C{\dot{V}}_{O2}$ and their relationship is described in Equation (7)
(8)$$C{\dot{V}}_{O2}=x\;C{\dot{V}}_{O2}+(1-x)\cdot C{\dot{V}}_{O2}$$

Equations (8)–(10) describe the systemic (S_.U._ and S_.L._) and pulmonary (P) oxygen equilibrium. Equation (8) shows that the oxygen flow rate in the upper systemic circulation ($C_{aO2}\cdot Q_{U})$ is reduced by the oxygen consumption of the upper body ($x\cdot C{\dot{V}}_{O2}$) leaving the reduced oxygen flow rate returning to the PA through SVC, where $C_{SVO2}$ is the systemic oxygen content of the venous flow.
(9)$$C_{aO2}\cdot Q_{U\;}-x\cdot C{\dot{V}}_{O2}=C_{SVO2}\cdot Q_{U}$$

Similarly, Equation (9), shows the oxygen flow rate into the S_.L._ which is a product of arterial oxygen content of blood, i.e., $\;C_{aO2}$ (in ml O_2_/_mL_ of blood) and $Q_{L\;}$ blood flow in the lower body and returning to the atrium. The total lower systemic oxygen flow rate is reduced by the oxygen consumption ($\left(1-x\right)\cdot C{\dot{V}}_{O2}$) in the S_.L._, leaving the reduced oxygen flow rate ($C_{SVO2}\cdot Q_{L}$) returning to the RA through IVC. Here $\left(1-x\right)$ term represents the fraction of the whole-body oxygen consumption consumed by the S._L_.
(10)$$C_{aO2}\cdot Q_{L}-\left(1-x\right)\cdot C{\dot{V}}_{O2}=C_{SVO2}\cdot Q_{L}$$

As we know in the HCSII procedure the blood flow through Pas, i.e., *Q_p_* is equal to the blood flow through S_.U._, i.e., Q_.U_. In Equation (10), the oxygen flow rate returning to the PA from the S_.U._ with the oxygen uptake in the lungs provides the oxygen flow rate returning to the atrium from the pulmonary circulation.
(11)$$C_{SVO2}\cdot Q_{U\;}+S{\dot{V}}_{O2}=C_{PVO2}\cdot Q_{P}$$

Equation (11) is used to track systemic oxygen saturation, which involves a complex function of cardiac output, the pulmonary venous oxygen concentration in the blood, lower body oxygen consumption, and ratio of blood flow between upper and lower systemic circulation.
(12)$$C_{aO2}\cdot \mathrm{CO}=C_{PVO2\;}\cdot \;\mathrm{CO}.-\left(\frac{Q_{L}}{Q_{U}\;}+1\right)\;(1-x)\cdot C{\dot{V}}_{O2}$$

Oxygen consumption has been determined based on literature-derived per-weight oxygen consumption 9 (mL/s)/Kg. The above expressions require cycle-average flow rate inputs originating from in-silico and in-vitro measurements (CO, Q_s_, and *Q_p_*), as well as literature-derived blood oxygen capacity and oxygen consumption data [31]. Pulmonary venous oxygen concentration was calculated from an assumed oxygen saturation (100, 95, 90, 85, and 80%) depending on patient ventilation and a given oxygen capacity (0.22 mL O_2_/mL of Blood). The oxygen saturation anywhere in the model can then be computed as $\frac{\mathrm{Oxygen}\;\mathrm{Concentration}}{\mathrm{Oxygen}\;\mathrm{Capacity}}.100$.

### 2.6. PowerLoss

Significant power loss across a domain can strongly indicate disturbed flow that can be correlated to excessive pressure gradients or momentum loss [27,40,51,52]. Power loss (*PL*) was estimated by first evaluating the flow field power at discrete locations. In this study, we define the Power loss and flow rate (*Q*) at a cross-sectional plane as
(13)$$Power=\left(\frac{1}{2}\rho \vec{v^{2}}+P_{static}\right)Q$$
(14)Q=VA

Power loss can then be evaluated as
(15)$$PL=P_{inlet}-P_{Outlet}$$
where, ρ is the density, V is the velocity, P is the static pressure, A is the conduit cross-sectional area. To normalize the power loss due to the baffle flow obstruction, the flow field power for models with the stented baffle $P_{baffle}$, is ratioed to the flow field power for a model with no obstruction $P_{no\;baffle}$ (Equation (14)).
(16)$$Efficiency\;\left(\eta \right)=\frac{P_{baffle}}{P_{no\;baffle}}$$

In measuring *PL*, the inlet was the pulmonary artery, and the outlet was the descending aorta. The *PL* was calculated as a function of the upper-to-lower body systemic flow distribution ratio i.e., 60–40 and 50–50, and as a function of the MPA narrowing.

### 2.7. In-Silico Particle Residence Time

Particle residence time (PRT) can help identify the presence of regions of recirculation, stagnation, or otherwise poor flow that can increase the risk of thrombus formation. Higher residence time in a specific region may cause platelets to accumulate shear stress and become activated [53,54,55,56,57]. PRT was calculated by releasing massless particles into the domain and tracking their residence time. Reininger et al. [53] calculated residence time to determine the effect of shear stress and residence time on fibrin clot formation in a laminar and turbulent flow. The study found that residence time is more critical for clot formation than the shear rate [57]. In similar studies, PRT calculations were different between patients and demonstrated the importance of PRT in identifying the recirculation and stagnation zones [55].

This study implemented a Lagrangian scheme to track these massless particles and measure PRT. After obtaining a converged flow field, the Lagrangian tracking scheme was activated, and particles were injected. Particles were assumed to be spherical and were released randomly in space and time for several cardiac cycles using an injection grid at the MPA inlet surface (Figure 13).

Particles were released with zero injection velocity at random across the grid (in space) and at random across the cycle (in time). The release of particles from each node of the grid was controlled by the point inclusion probability function built-in StarCCM+, which regulates which nodes on the injection grid release a particle at any given time step.

### 2.8. Experimentation

An efficient MFL tuning process plays a crucial role in achieving accurate experimental results. MFL setups can be devised by following various LPM to perform different kinds of experiments [32]. As this MFL setup involved various components for replicating the reduced LPM as described in Section 2.2, each component was carefully calibrated to minimize imprecision. The MFL setup has been systematically tuned by following a sequential “Bottom-Up” procedure, as explained in our previous study [31].

The benchtop experimentation of HCSII was conducted to match catheter tracings and clinical reports of two patients with different CO i.e., patient-1 with CO of ~2.99 L/min and patient-2 with CO of ~1.75 L/min. As mentioned in Section 2.2.2, the MFL setup consisted of four compartments, i.e., upper systemic, lower systemic, left pulmonary, and right pulmonary circulations. The 3-D printed phantom representative of the reconstructed HCSII physiology described in Section 2.1.2 has been integrated with the physical components replicating the R, and C parameters of the reduced LPM, which were derived from the clinical measurements. Figure 14a shows the complete MFL setup of the HCSII circulation. The main objective of this in-vitro study was to investigate the hemodynamics in the reconstructed anatomy of the HCSII circulation; hence, VSD and the BDG procedure were modeled in the LPM as a peripheral component to this simulated surgical procedure. Two resistor valves (i.e., R__MPA_ and R__Asc. Aorta_) were used to represent the VSD feature in the MFL physically, and the BDG procedure was replicated using a “T” junction, as shown in Figure 14a and Figure 15a. For in-vitro measurements, the blood was modeled as Newtonian. A custom working fluid (ρ = ~1000 kg/m^3^) was developed to conduct the benchtop experiments. This batch of fluid was developed by mixing 56% of Glycerin and 44% of water to match the viscosity of the blood, i.e., four centipoises.

The MFL was driven by the Harvard Apparatus medical pump replicating the viable RV in the HCSII anatomy. This positive displacement piston pump was tuned to produce the pulsatile cardiac waveform matching the CO respective patients, as mentioned in the catheter reports. Further, a beats per minute (BPM) decal in the pump was adjusted to generate the heart rate (HR) of 80 BPM for a suitable stroke volume value calculated from the clinical data provided by our clinical partners. Figure 14b shows the location of resistors, compliance chambers, flow, and pressure sensors in all four compartments of the MFL. The table mentioned in Appendix B lists the names of the elements (including measuring and sensing devices) with their corresponding location in the MFL setup. Each compartment of the MFL was constructed with four instruments (i.e., a needle valve, a compliance chamber, a flow rate sensing device, and a pressure sensing device) which were placed sequentially, as shown in Figure 14b and Figure 15b. For both patient cases, the upper and lower systemic resistors were tuned to maintain the flow distribution between the upper and lower systemic circulation to match the respective catheter data. Then, the pulmonary resistances were tuned so that the flow returning from S_.U._ to PAs through SVC remained equally distributed on each side, and the ratio of pulmonary to systemic flow was maintained.

As one of the critical objectives of this study was to experimentally investigate the hemodynamics in the reconstructed geometry of the HCSII circulation, we integrated the pressure sensors in the specific locations of the MPA, AA, and DA to probe the hemodynamic pressure field transiently over cardiac cycles. Figure 15a shows the integration of the sensors and the resistors to the 3-D phantom in the MFL. Once the correct flow distributions between each LPM compartment were achieved by tuning these resistances, then all the compliance chambers are re-engaged. Figure 15b shows all four compartments of the MFL. Pressures and flow rates in the MPA, AA, and DA were maintained by tuning the respective resistances. These integrated pressure sensors collect pressure from the MPA section over the baffle, the AA, and DA.

In this way, clinically correct hemodynamics was achieved in these compartments of the MFL. An in-house LabVIEW code has been developed to visualize and acquire the hemodynamic data from the MFL setup. All the integrated digital flow sensors (FMG90 series OMEGA Engineering, Norwalk, CT, USA) and analog pressure sensors (PX315-015GV series, OMEGA Engineering) in the MFL were connected to National Instruments (NI, Austin, TX, USA) data acquisition board (DAQ), i.e., NI- CompactDAQ-9174 containing an analog module card NI DAQ 9205 and a digital module card NI DAQ 9361 respectively. The transient and cycle averaged hemodynamic (pressure and flow rate) waveforms acquired from each integrated sensor in the MFL were visualized through the LabVIEW code. While acquiring the data during experimentation, noise interference is observed. The trend of noise interference has been consistent for both experimental runs (patient-1 and patient-2). A low-pass Chebyshev type-2 filter was designed to filter the noise. To keep the brevity, the filtered hemodynamic waveforms acquired at key physiological locations in the MFL setup obtained for both (patient-1 and patient-2) experimental runs are shown in Figure 16 and Figure 17.

To conduct the in-vitro PRT experiment, a high-speed SONY DSC-RX10 III Cybershot camera was installed in the MFL by calculating the hyperfocal distance between the camera lens and the MPA conduit present in the 3-D phantom. Two non-rectified light sources have been installed on MFL by focusing on the MPA conduit in the HCSII phantom.

#### 2.8.1. In-Vitro Particle Residence Time

As stated above, in-vitro particle tracking is a critical parameter in determining the pathological flow conditions in the proposed procedure. The presence of regions of stagnation and recirculation zones leads to higher PRT values. These longer PRT values are an indicator that shear stress is being accumulated by platelets leading to the activation of platelets, and thus, thrombus formation [54].

In this in-vitro study, we have implemented an experimental technique to track injected particles in the MPA conduit. Once the converged flow field was achieved in the respective LPM compartments, then particles were injected into the MPA conduit through the atrium chamber. To conduct in-vitro PRT experimentation, we have used fish eggs (CAVIAR RUSSE American Sturgeon) of 2 mm diameter to replicate the massless particles as modeled in the in-silico study (Figure 18a). The density of this specific caviar grade (ρ = 1010 kg/m^3^) closely matched the density of the custom fluid developed for performing the benchtop experiments. To inject the particles randomly with zero velocity, as implemented in the in-silico domain, we have mixed the particles (by random volume) in the atrium chamber of the MFL, which were pumped in the MFL by the Harvard apparatus pump and injected into the MPA conduit during the systole period.

We have implemented a computer vision technique to track the trajectories of each injected particle in the MPA conduit. This technique involved the high-speed imaging of the flow field in the MPA conduit (Figure 18b) utilizing a high-speed (HS) SONY DSC-RX10 III Cybershot camera (SONY, Tokyo, Japan). This HS camera has imaged the flow field with an acquisition rate of 960 frames per second.

We developed an in-house multi-objective Kalman filter-based motion tracking scheme using MATLAB R2021a (MathWorks, Natick, MA, USA) to post-process this video (Figure 19). The acquired video streams needed to be stabilized due to the non-rectified light source and mechanical vibration of the MFL setup. An FFmpeg platform-based in-house algorithm was used as a preprocessor to stabilize the acquired video streams. The tracking algorithm used Matlab software-based inbuilt “Foreground subtraction” and “blob detection” schemes to detect the pixels on an 8-bit binary mask. Finally, the Kalman filter was implemented on the detected blob (per frame) to track each detected particle’s trajectory for the acquired set of frames. Kalman filter was trained using an adaptive learning scheme. The Munkres’ assignment algorithm governed the task of assigning detection to a track.

#### 2.8.2. Camera Calibration

Computer vision algorithms have been used to perform in-vitro PRT analyses, which involved video imaging of the flow field in the MPA conduit using a high-speed SONY DSC-RX10 III Cybershot camera. Camera resectioning is the primary step for performing video imaging of any dynamic event. Using a full camera calibration algorithm, we have estimated the parameters of this high-speed camera’s lens and image sensor. These parameters were used to correct the lens distortion and identify the location of the object and the camera in the world coordinate system (WC) (Figure 18b). We have used the Computer Vision Toolbox of MATLAB software (MathWorks, Natick, MA, USA) for calculating the intrinsic, extrinsic, and lens distortion coefficient in the camera parameters. This model includes the pinhole camera model with radial and tangential lens distortion [58,59,60].

To perform the camera resectioning process in the MFL setup, we have used three sets of checkerboard patterns with details mentioned in Table 5 and shown in Figure 18a and Appendix A. We took multiple images (~10 images in each subset) of each calibration set from different angles. The pattern was placed precisely at the same location where the 3D phantom was placed in the MFL setup.

Among all calibration data sets used for the calibration process, set-2 generated the intrinsic and extrinsic parameters with the least errors. Hence, we conducted the PRT analyses using the camera matrix parameters obtained from the calibration set-2. Also, we have performed the uncertainty quantification and error analyses for each calibration set. The overall mean projection errors in each calibration, i.e., set-1, set-2, and set -3, are 2.40 pixels, 2.26 pixels, and 10.46 pixels, respectively, as mentioned in Table 5.

### 2.9. Statistical Analysis

As mentioned in the previous sections, in-vitro measurements may involve a certain amount of instrumentation error and uncertainties. In our current study, we have performed two statistical analyses: Paired T-test and One-way ANOVA to statistically quantify the reliability and significance of the in-vitro findings. A test is considered statistically significant (significantly different) if and only when the H_0_ is rejected. The *p*-value is often used in replacement for the critical region to accept or reject the null hypothesis. The *p*-value for the given test stat Y(X) is given as by
(17)$$\mathrm{two}\;\mathrm{sided}:\;PH0\left(\left|Y\right|\geq y\left(x\right)\right)=p-value$$

For the null hypothesis to be rejected, the *p*-value should be smaller than the significance level (α). The *p*-value is a random variable that follows a normal distribution under the null hypothesis. In this study, we have conducted the following types of statistical analyses:Paired T-test

This test was conducted to statistically verify if the in-vitro measurements closely replicate the hemodynamic findings of the catheter data.

**H_0_:** The cycle averaged in-vitro measurements are closely replicating the hemodynamic observation mentioned in the clinical reports, i.e., without showing statistically significant deviation (p > 0.05) from the clinical reports.

**H_1_:** The cycle averaged in-vitro measurements are statistically significantly different than the hemodynamic observation (p < 0.05) mentioned in the clinical reports.

2.One-way ANOVA

This test was conducted to statistically confirm if the individual PRT measurements for each patient in in-vitro studies were statistically different from each other.

**H_0_:** The transient PRT measurements for the same patient in the in-vitro studies were closely replicating the same trends, i.e., without showing significant deviation (p > 0.05).

**H_1_:** The transient PRT measurements for the same patient in the in-vitro studies were showing significantly different trends (p < 0.05).

## 3. Results and Discussion

### 3.1. HCSII Hemodynamics

In-vitro and in-silico models were tuned to match catheter tracings and clinical reports provided by our clinical partners from APH. Data for two patients were made available, patient 1 with a higher CO of 2.99 L/min and patient 2 with a lower CO of 1.76 L/min. From Table 6, Table 7, Table 8 and Table 9, it can be observed that cycled-averaged hemodynamic data from in-vitro and in-silico measurements for flow and pressure waveforms compared well to the catheter data following the HCSII procedure. This would indicate the tuning process for both in-vitro and in-silico models has been carried out successfully. While we compare the catheter tracings with the in-vitro and in-silico findings, it can be observed that in most locations, the values reported for the flow rate measurements are less than seven percent and for the pressure measurements are less than five percent. The maximum deviation between in-silico and in-vitro findings can be found in the DA is approximately eight percent.

Statistical analyses were performed on the simulated hemodynamic measurements to estimate how closely the multiscale computational and experimental observations follow the trends of the catheter reports for both patients. Table 10 and Table 11 summarize the two-tailed paired T-test analyses findings for cycle-averaged hemodynamic variables, i.e., flow rate and pressure obtained from simulated experiments (i.e., CFD and MFL runs) and patient-specific catheter reports. From the statistical results mentioned in Table 10 and Table 11, it can be observed that all *p*-values confirm the initial (i.e., null) hypothesis H_0_ (*p* > 0.05) with a Pearson Product Moment correlation coefficient (r) = 0.9, which signifies that the in-vitro and in-silico models are strongly correlated and both multiscale CFD and MFL models have been tuned successfully.

Further comparisons of the waveforms between catheter tracings and in-vitro pressure measurements reveal closely matching trends, as shown in Figure 20 and Figure 21. Peak systolic, diastolic pressures, and pulse pressures match accurately. It is important to note the difference in time periods of the pulsatile hemodynamic waveform in the patient data and the simulated in-vitro results. The HR from pressure tracings was estimated to be 120 beats per minute (bpm). This rate was replicated in the in-silico simulations. However, in MFL, the HR was set to 80 bpm due to operational limitations of the pump (acting as RV in the MFL). Despite this drawback, the pump settings were manipulated (for decreased bpm, the stroke volume was increased) to ensure that physiological conditions were consistent across in-vitro and in-silico investigation techniques.

### 3.2. Particle Residence Time (PRT)

#### 3.2.1. In-silico Study

Massless particles were injected throughout the cardiac cycle (Figure 22). To quantify the residence time each fluid particle took to travel from the MPA to the DA across the baffle for each geometrical variation, we estimated PRT. Those were traced to measure the total PRT for each particle released and then calculated the average PRT over all the particles injected (Figure 22 and Figure 23).

The results show that particles take less than ~0.5 s to exit the fluid domain (0.5 s being a single heart cycle). However, it must be noted that some particles took longer to flow downstream (≥3 s), as shown in Figure 23.

This result means that most particles flushed out of the computational domain during each heart cycle; hence the baffle did not cause significant pathological flow patterns for the models we investigated. A periodic aspect of the average PRT can be observed in Figure 23, which reflects how particles were released randomly across the cycle, as their transport varies according to the cycle flow field features.

#### 3.2.2. In-vitro Study

Similarly, in the in-vitro domain, we measured the time the injected particles took to travel from the MPA inlet to the DA outlet. Each particle’s trajectory was tracked throughout the MPA conduit during various phases of the cardiac cycle using the kalman tracker-based multi-objective detection algorithm, as shown in Figure 24.

Kalman filter-based object detection and tracking algorithm is a stochastic process. This type of algorithm involves stochastic uncertainty and errors. We have performed multiple experiments of tracking the particles for each case (patient-1 and patient-2) to minimize the errors associated with the particle tracking algorithm and accurately calculate the average residence time for both patient cases. Table 12 and Table 13 elucidate the particle residence time for a random number of particles tracked in each cardiac cycle. To compute the PRT for patients 1 and 2, we conducted five and three MFL-based experiments of tracking particles for six and five consecutive cardiac cycles, respectively. Utilizing our in-house Kalman filter-based multi-object tracking algorithm, we have tracked the trajectories and derived velocities of each particle entering the MPA conduit till it is flushed out through DA to the peripheral circulation during every cardiac cycle. Every PRT experiment has been performed for six and five consecutive cardiac cycles to calculate the cycle averaged residence time for patient-1 and patient-2 cases, respectively. From Table 12 and Table 13, it can be inferred that, on average, the particle residence time is shorter than the respective cardiac cycle. Some experimental cases show deviation from the data’s central tendency due to the modeling error and measurement error associated with the implementation of the Kalman filter algorithm.

We performed statistical analysis using one-way ANOVA to verify if all the PRT experiments conducted for each patient case belonged to the same population. Statistical reports (Appendix A) show that the null hypothesis H_0_ cannot be rejected (p > 0.05) for the data obtained from the PRT experiments. Hence it can be inferred that all the experiments for each patient case (listed in Table 12 and Table 13) belonged to the same population, and the difference between each experimental group was statistically insignificant.

To better display the variation in PRT, Figure 25 and Figure 26 were generated. These plots offer insight on the average PRT across several cardiac cycles revealing the deviations in particle transport. The expected value for the particle’s transport across the lower circulation domain could be compared to the cardiac cycle period (i.e., 0.75 s); higher average values would indicate the presence of pathological flow patterns (e.g., presence of recirculation zones in the flow field). As shown in Figure 25 and Figure 26, the measured PRT varied as high four-fold the cycle period indicating that particles were caught in the recirculation zones either in the enlarged MPA or distal to the baffle.

Furthermore, we have plotted the flow field (velocity vector plots) and trajectories of the tracked particles in the MPA conduit for both patient cases that were obtained after performing in-vitro PRT study, as shown in Figure 27, Figure 28, Figure 29 and Figure 30, Figure A1 and Figure A2 and Appendix A. The presence of the void region in the top sections of the trajectory and velocity vector plots (Figure 27, Figure 28, Figure 29 and Figure 30), derived from the experimental PRT runs, show the obstruction to the field of view in the DA conduit of the experimental 3D phantom due to the presence of the pressure sensor integrated with the MPA conduit. Flow field (Figure 27, Figure 28, Figure 29 and Figure 30) was derived from the tracked particles using several consecutive heart cycles while experimentally simulating the hemodynamics for both patients. The flow field is cyclic; hence the flow patterns were repeated in every heart cycle. By performing a qualitative inspection, it can be found that there are two flow regimes found in two distinct regions:MPA to proximal of the bafflePosterior of the baffle.

In region (i), the flow regime is more disordered than in region (ii), suggesting flow stagnation and circulation due to the enlarged MPA and stented-baffle obstruction. In region (ii), the flow field seems organized and unidirectional due to more regular anatomy. As discussed by Hameed et al. in [23], the peak Reynold number estimated for this flow field was 1807, which is well within the laminar flow regime. In-vitro findings from our current study closely matched the in-silico observations reported in our previous work [23].

### 3.3. Power Loss

Power loss and flow field efficiency were calculated with respect to MPA narrowing. As aforementioned, previous studies did not identify significant pressure gradients across the baffle. Similarly, the current measurements determined that the PL and the efficiency reflect the absence of excessive obstruction. The power loss found in this study is due to baffle-related obstruction of the MPA, which causes a pressure drop and viscous losses. The average power loss and efficiency related to the systemic flow distribution ratio variation were 11.50 ± 1.39 mW and 0.90, respectively (Table 14). As the lower body perfusion increases, the power loss grows. By maintaining a constant flow ratio (60–40) and varying MPA narrowing, the average power loss and efficiency were found to be 12.46 ± 0.18 mW and 0.91, respectively (Table 15).

Table 15 offers a closer look at the power loss and efficiency as the MPA narrowing changes. The first row of the results represents the measured quantities of a case where no baffle obstruction occurs. The measured PL was negligible in all models. The MPA-1 case showed a slight increase (about 2%), while in the MPA-2 case, PL increased by 5%. As the narrowing kept increasing, the PL did not change, as seen in the nominal case where PL was the same as what MPA-2 experienced. More drop-in PL was measured in the MPA-3 case, where the narrowing is larger than the nominal case. However, no change in PL was calculated in MPA-4, which has a large MPA narrowing. The efficiency, as defined above, was also calculated to quantify the effect of the baffle’s presence on the flow. The same trend was seen here, and the efficiency did not drop below 90% in all cases. The lowest value was calculated in MPA-2 and the nominal cases where we have the highest PL increase while the rest have the same efficiency (97%).

### 3.4. Oxygen Transport

#### 3.4.1. Effect of Ascending Aorta

The oxygen delivery was reported with different pulmonary venous saturation values (100%, 95%, 90%, 85%, 80%) depending on how well a patient is ventilated. The total cardiac output for this study was 2.93 ± 0.02 (L/min) with a 60–40 split ratio between the lower body and upper body circulation and nominal MPA narrowing, as shown in Table 16. The tabulated values reveal that alterations to the AA size do not significantly alter flow distribution to the lower body (no higher than 1%). On the other hand, the upper circulation sees a significant change in the flow distribution. Coronary arteries are observed to have the least change in the flow distribution (up to 3%). Carotids and Subclavians register the largest change in flow distribution (up to 37%).

As shown in Figure 31, changes in the AA diameter did not impact the oxygen saturation results compared to the nominal model, even though the upper flow levels were impacted in some cases. The saturation rates for the whole system as the AA diameter increased did not experience a significant impact, about 1% only. The measured oxygen values fall in the range of reported clinical rates for the BDG procedure.

#### 3.4.2. Effect of MPA

Hameed et al. in [23] reported that no relevant average pressure drops across the baffle were observed based on a clinically suggested threshold for the range of the MPA narrowing. In this study, we looked into oxygen transport for the same synthetic geometries to quantify the effects of MPA narrowing on oxygen delivery (Figure 32). In Table 17, we reported the flow distribution with a 60/40 split ratio as a function of the MPA narrowing. The tabulated values describe that alterations to the MPA narrowing size do not significantly alter flow distribution to the lower body (no less than 2%). As opposed to the observations made in Table 16, the upper circulation sees a smaller change in the flow distribution. Coronary arteries are observed to have a change in the flow distribution of up to 3%. Carotids and Subclavian arteries register a larger change in flow distribution as coronaries of up to 9%.

Calculated oxygen saturations did not reveal any changes as the MPA narrowing changes (Figure 32). This lack of response was expected for two reasons,

The oxygen transport is a function of systemic flow distribution ratio and changes significantly as the systemic flow distribution ratio changes.The MPA narrowing, as previously reported, does not cause large pressure gradients, significantly altering flow distribution.

#### 3.4.3. In-Vitro Study

As outlined in the introduction, the HCSII circulation features an upper circulation that traverses the lungs, while the lower body circulation bypasses the lungs and directly feeds back to the RA. This physiology causes the systemic flow to mix oxygenated and deoxygenated blood. Hence it becomes crucial to track systemic oxygen saturation. These oxygen saturation values reported in in-vitro experiments were calculated using the oxygen transport model employed in the CFD instead using the flow values in the flow loop. In the following plots, systemic saturation for the in-vitro study is reported for a range of pulmonary venous saturations (80–100%), shown in Figure 33. It can be observed that for healthy levels of pulmonary venous saturations (95–100%), systemic saturations were maintained within an acceptable clinical range (>85%). As the pulmonary venous saturation dropped below 95% (considered pathological), the systemic saturation was no longer viable as expected. These observations hold true for both patients.

Oxygen transport calculations for the in-silico study yielded similar results for MPA narrowing and AA diameter alterations. The trend associated with the reduction in pulmonary venous saturation is retained.

## 4. Conclusions

To this end, this study demonstrated computational and experimental methodologies to quantify and qualify the detailed hemodynamics of the novel surgical techniques, i.e., HCSII. The multiscale in-silico and in-vitro investigations allowed us to determine the potential presence of any pathological flow fields and estimate the efficacy of tools implemented to improve patient care. In-vitro and in-silico techniques were utilized in this study to investigate the effects of aortic root size, main pulmonary artery narrowing, alteration on oxygen transport, and flow distribution in this novel procedure.

Based on the clinical data, we carried out cross-validations for the in-vitro and in-silico results. Firstly, tabulated cycled average results reveal a high degree of similarity to catheter data. Secondly, a comparison of waveforms revealed also matching trends in peak systolic values and pulse pressures. Thirdly, the statistical analysis comparing patient data to in-silico and in-vitro studies utilizing paired *t*-tests and one-way ANOVA showed a high level of agreement among all techniques implemented.

Oxygen observations reveal that despite blood mixing leading to systemic saturations for healthy individuals with pulmonary venous saturations in the range of 95–100, Systemic saturations were clinically viable regardless of MPA narrowing or AA size.

Based on the particle residence time study, particles were found to be flushed out of the domain within a single heart cycle. No pathological flows were observed. Particle tracking carried out in-vitro study seems to show two flow regimes from the MPA to DA

Seeming disturbed flow (MPA to proximal of the baffle)Organized flow (Posterior of the baffle to DA)

## 5. Future Works

In the present study, although the vessel walls of the MPA and AA conduits of the synthetic HCSII phantoms have been assumed as rigid, this assumption may affect the evolved flow field locally in those conduits. However, we have precisely replicated vascular (arterial and venous) compliances in each LPM compartment to accurately replicate the real-time response of the peripheral circulation coupled with the synthetic HCSII domain. In the computational PRT study, we assumed massless particles passively convected by the converged fluid domain. The adhesion between the particles and the walls was not modeled.

In the in-vitro study, the MFL setup did not replicate the autoregulatory feedback response mechanism of the cardiovascular system. In our future studies, we plan to involve a state-of-the-art controls system and sensor fusion technology to develop an automated MFL setup based on the Hardware-in-the-loop technique such that our benchtop setup can replicate such autoregulatory responses of the circulatory system. In addition, in the experimental PRT study, only the particles with a size of 2 mm in diameter were injected into the MPA conduit. In the future, different size particles with varying masses will be injected into the MPA conduit to perform extensive PRT analyses. Also, to make the tracking technique more robust, the extended Kalman filter or unscented Kalman filter algorithm will be implemented.

## Figures and Tables

**Figure 1 bioengineering-10-00135-f001:**
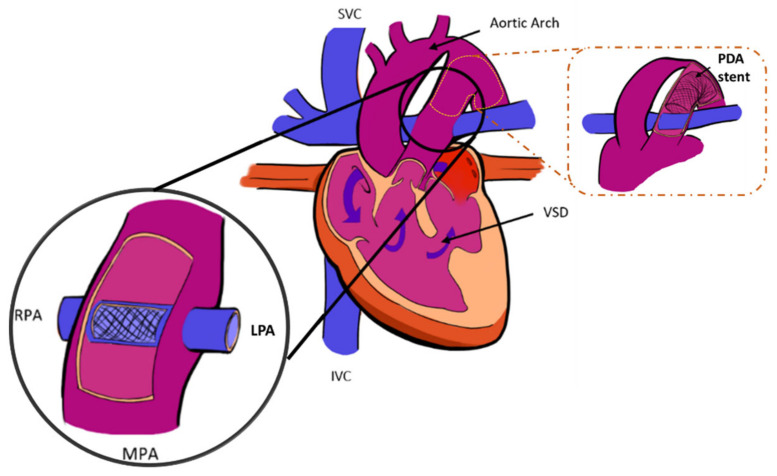
Schematic of the HCSII procedure [22] (p. 329) (MPA = main pulmonary artery, SVC = superior vena cava, IVC = inferior vena cava, LPA = left pulmonary artery, RPA = right pulmonary artery, VSD = ventricular septal defect, PDA stent = patent ductus arteriosus).

**Figure 2 bioengineering-10-00135-f002:**
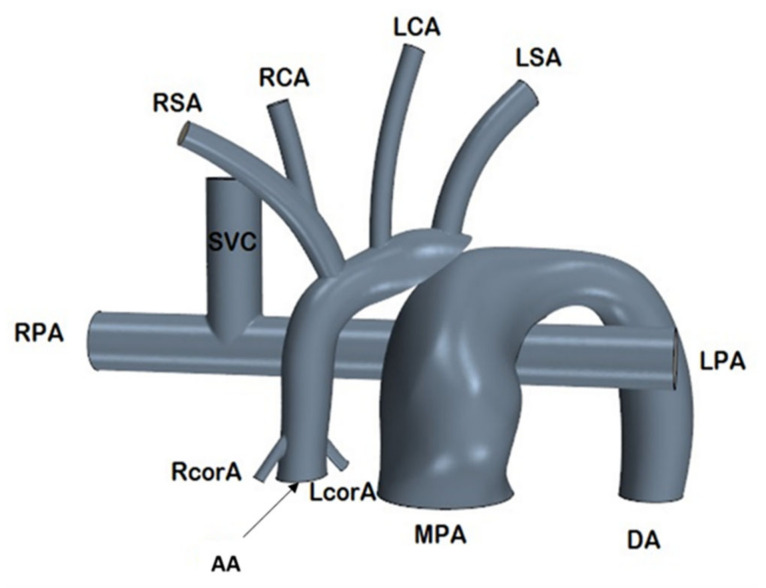
The synthetic HCSII geometry used in in-silico studies.

**Figure 3 bioengineering-10-00135-f003:**
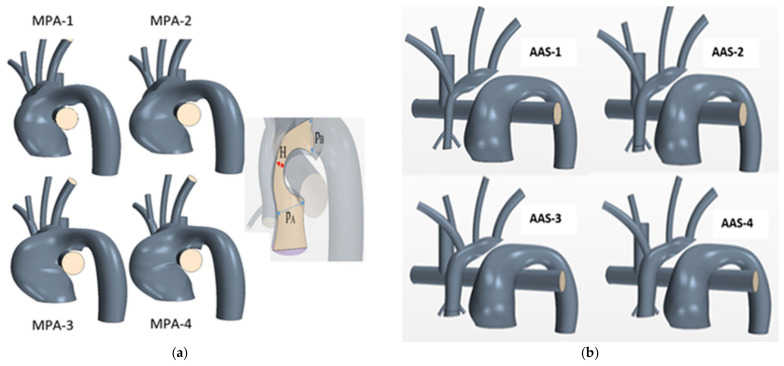
Patient generic synthetic geometries for varying (**a**) MPA narrowing denoted as H, and (**b**) ascending aorta root diameters ($D_{AA}$).

**Figure 4 bioengineering-10-00135-f004:**
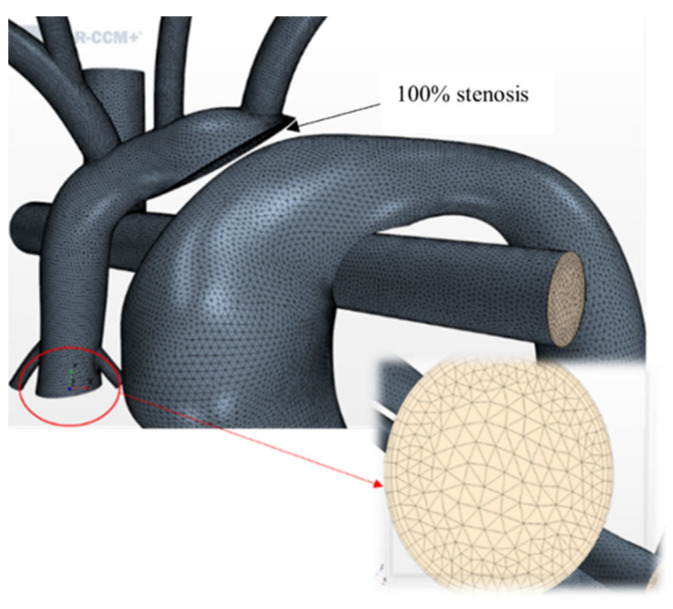
Meshed HCSII geometry with a close-up of the ascending aorta inlet boundary.

**Figure 5 bioengineering-10-00135-f005:**
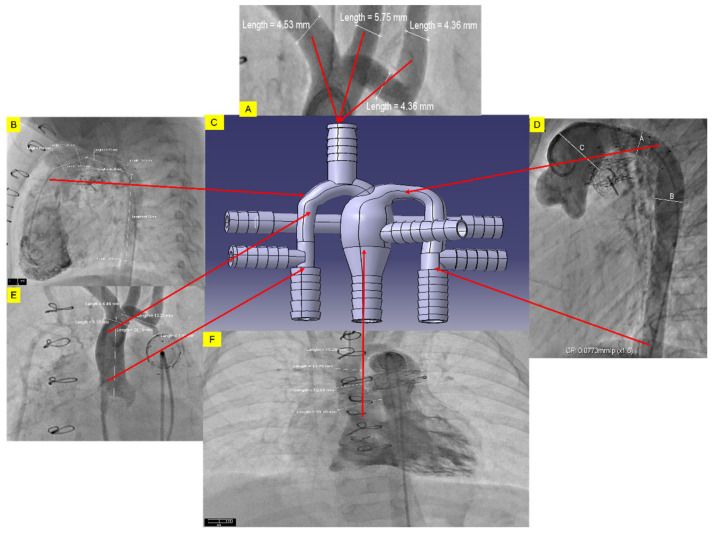
Development of a 3D CAD model of the HCSII centerpiece from deidentified angiographic images (**A**). Diameter of RCA, RSA, LCA, and LSA (**B**). Diameter of MPA root (**C**). Dimension of the 3D phantom with the labeled diameters (in mm) of labeled branches using CATIAv.5 (**D**). Diameters of the aortic arch and descending aorta (**E**). Diameter of AA (**F**). Diameter of MPA.

**Figure 6 bioengineering-10-00135-f006:**
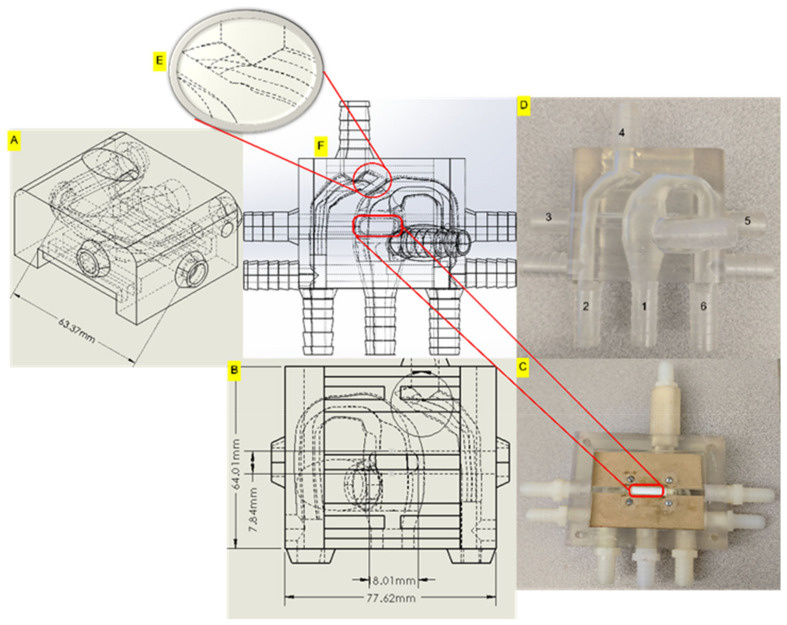
(**A**). Side view of the top half, (**B**). Anterior view of the centerpiece with rails (**C**). Bottom half of the centerpiece with stent and gasket. Interior view of the centerpiece with rails, (**D**). 3D phantom with the labeled conduits (**E**). Completely stenosed section in the HCSII anatomy. (**F**). HCSII centerpiece within the rectangular block.

**Figure 7 bioengineering-10-00135-f007:**
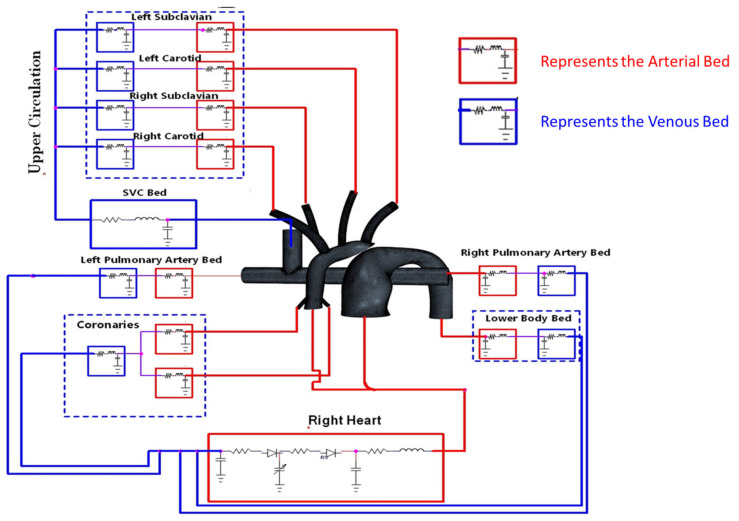
Multiscale model of the HCSII circulation, three-dimensional CFD model coupled with the lumped parameter model.

**Figure 8 bioengineering-10-00135-f008:**
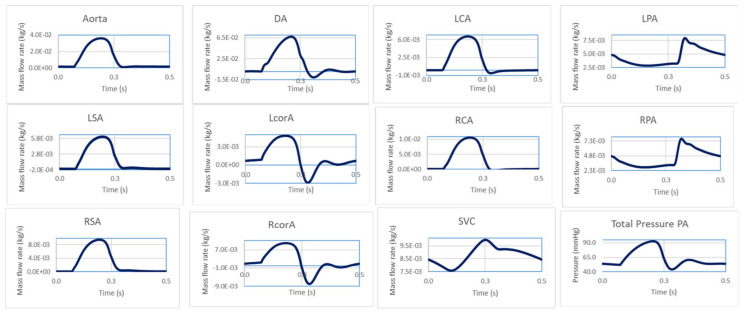
Sample BC waveforms for all inlets and outlets used in the CFD domain.

**Figure 9 bioengineering-10-00135-f009:**
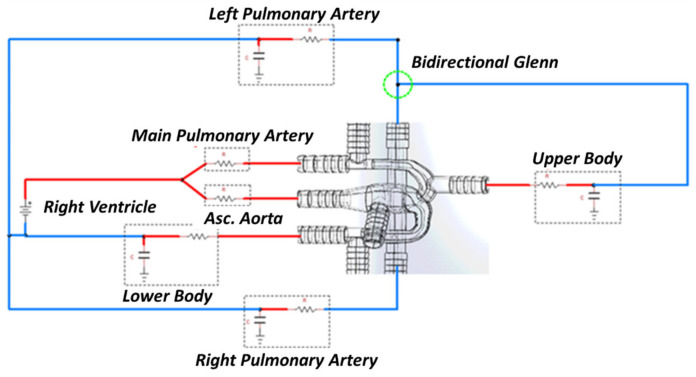
Reduced LPM representation of HCSII MFL setup.

**Figure 10 bioengineering-10-00135-f010:**
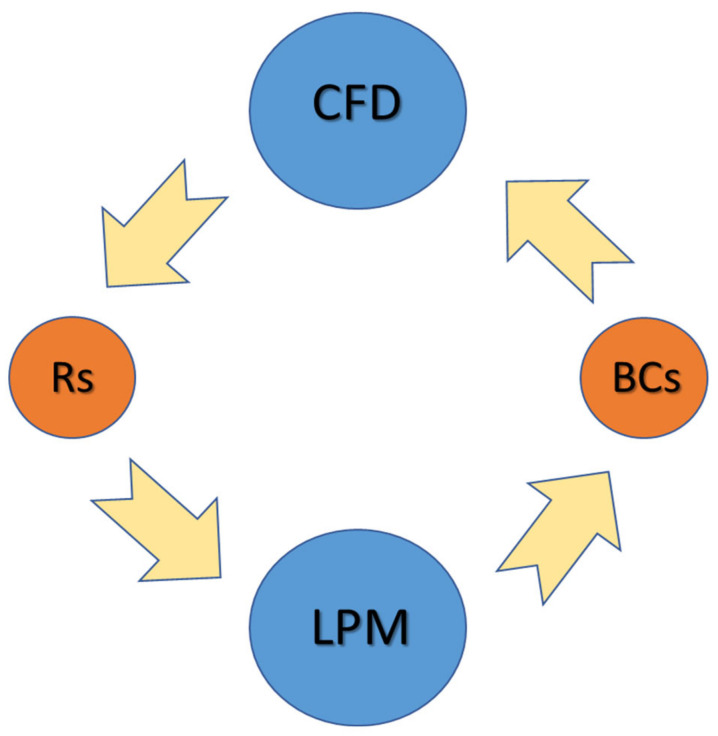
3D CFD to 0D LPM coupling scheme.

**Figure 11 bioengineering-10-00135-f011:**
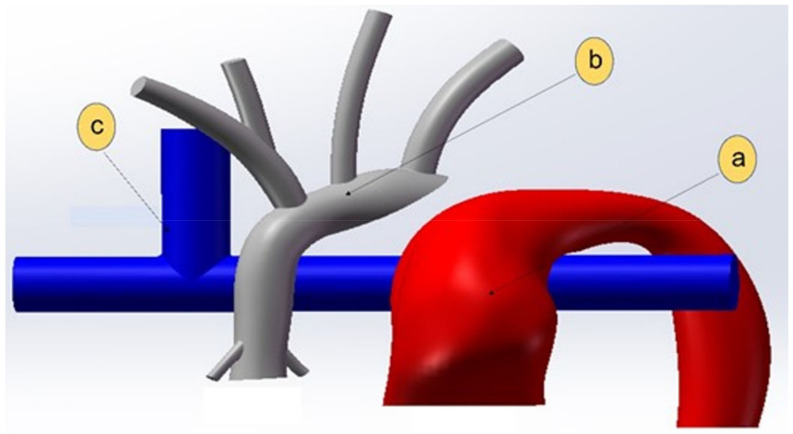
Fluid domain regions: (**a**) lower body circulation shown in red, (**b**) upper body circulation in grey, and (**c**) the pulmonary circulation shown in blue.

**Figure 12 bioengineering-10-00135-f012:**
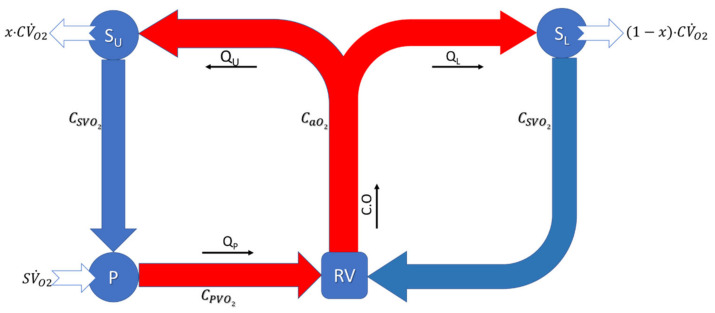
Generic oxygen transport model schematic of HCSII circulation (RV = Right Ventricle, S_U_ = Upper Systemic circulation, S_L_ = Lower systemic circulation, P = Pulmonary circulation, CO = cardiac output [L/min], Q_U_ = Upper body flow [L/min], Q_L_ = Lower body flow [L/min], C_xxO2_ = Oxygen concentration (mL O_2_/_mL_ of blood) for a = arterial; sv = systemic venous; pv = pulmonary venous, $C{\dot{V}}_{O2}$ = 9. m [mL/min], systemic oxygen consumption with m = patient weight [Kg] x = oxygen consumption split ratio, $S{\dot{V}}_{O2}$ = Oxygen uptake in the lungs.

**Figure 13 bioengineering-10-00135-f013:**
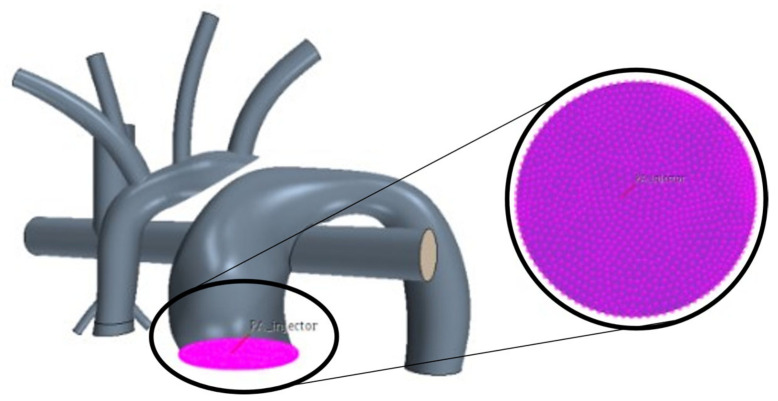
Injection grid located in the main pulmonary artery.

**Figure 14 bioengineering-10-00135-f014:**
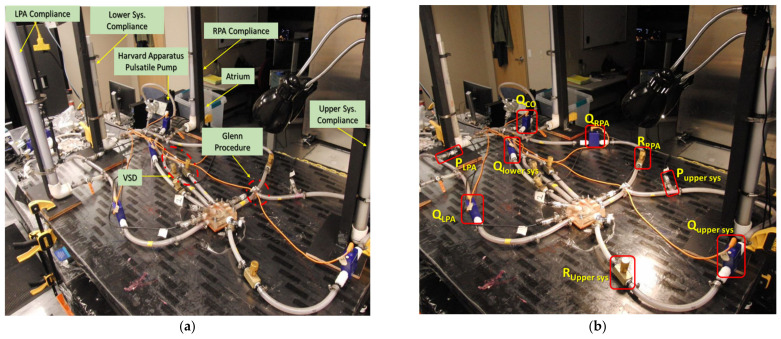
(**a**) MFL set up of HCSII circulation; (**b**) Position of sensing and measuring devices in the MFL setup.

**Figure 15 bioengineering-10-00135-f015:**
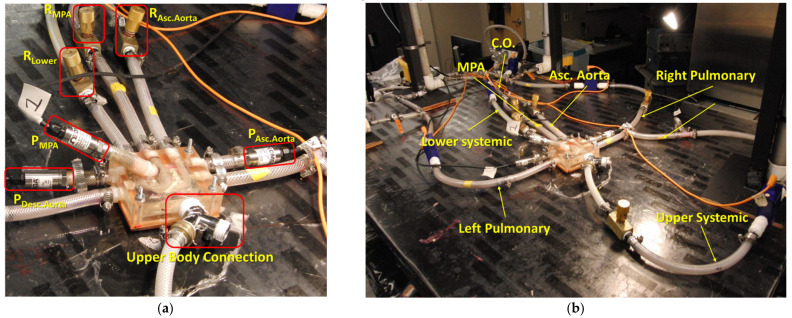
(**a**) 3-D phantom integration with the MFL; (**b**) Systemic and Pulmonary circulation in the MFL.

**Figure 16 bioengineering-10-00135-f016:**
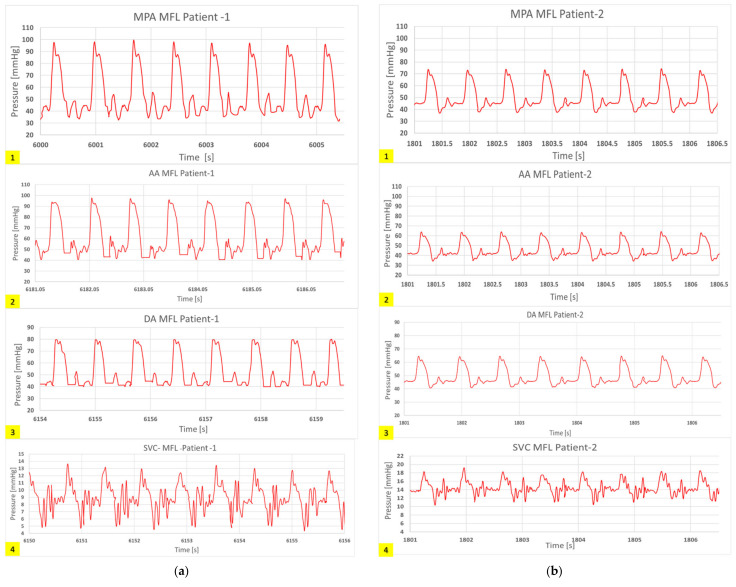
In-vitro results showing hemodynamic pressure waveforms (in mmHg) acquired at the following locations in the MFL are 1. MPA conduit in HCSII anatomy, 2. AA conduit in HCSII anatomy, 3. DA conduit in HCSII anatomy 4. SVC at the analogous BDG section, for (**a**) the experimental simulation of patient-1 case (CO = 2.99 [L/min]), (**b**) the experimental simulation of patient-2 case (CO = 1.75 [L/min]).

**Figure 17 bioengineering-10-00135-f017:**
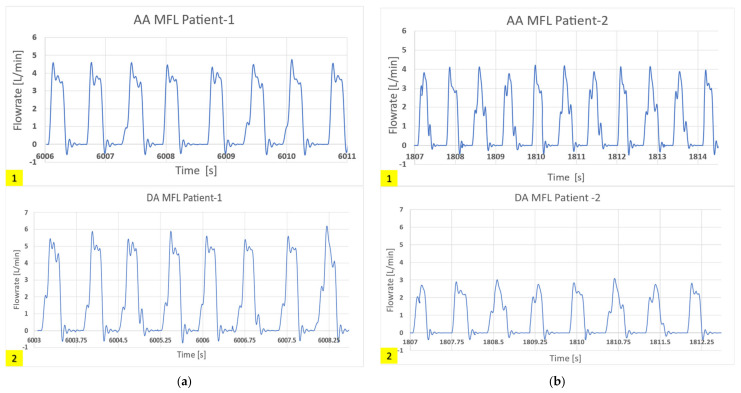
In-vitro results showing hemodynamic flow rate waveforms (in L/min) acquired at the following locations in the MFL are 1. AA conduit in HCSII anatomy, 2. DA conduit in HCSII anatomy for (**a**) the experimental simulation of patient-1 case (CO = 2.99 [L/min]), (**b**) the experimental simulation of patient-2 case (CO = 1.75 [L/min]).

**Figure 18 bioengineering-10-00135-f018:**
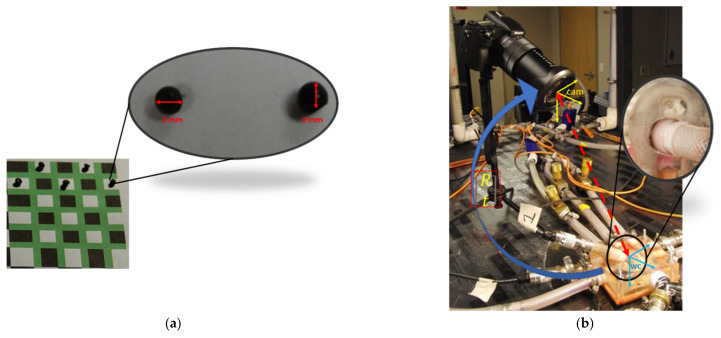
(**a**) A sample 5 × 6 Checkerboard image with the colored pattern used for calibrating the camera; (**b**) Camera resectioning process in the MFL setup by highlighting the region of interest, i.e., MPA conduit.

**Figure 19 bioengineering-10-00135-f019:**
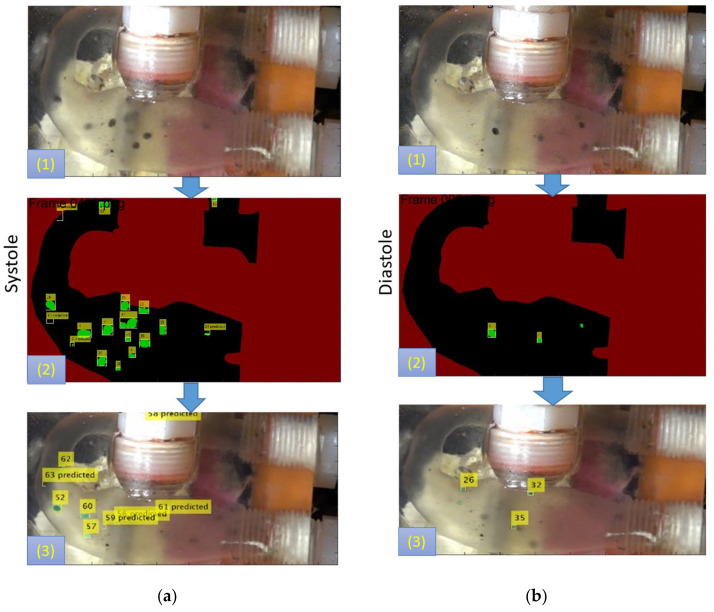
The figure shows the Foreground subtraction (step 1) with the blob detection (step 2) and the multi-objective Kalman filter-based tracking process (step 3) of the particles in the MPA conduit during the (**a**) systole and (**b**) diastole phase of the cardiac cycle.

**Figure 20 bioengineering-10-00135-f020:**
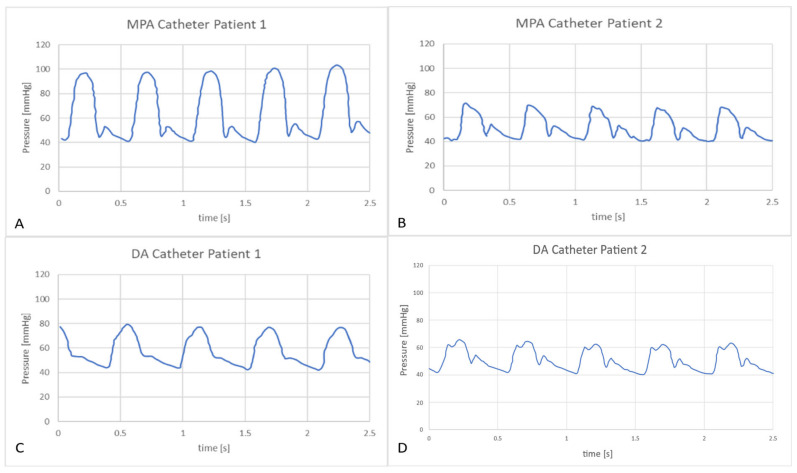
Catheter tracings of pressure waveforms in the MPA (**A**,**B**) and DA (**C**,**D**) conduits for patients 1 and 2 respectively.

**Figure 21 bioengineering-10-00135-f021:**
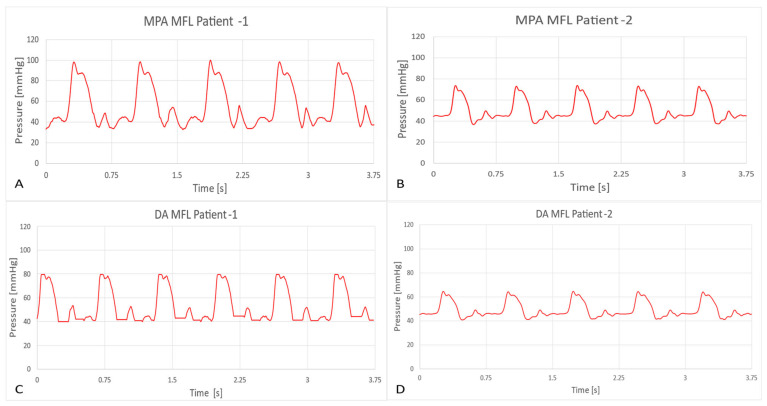
In-vitro measurements of pressure waveforms in the MPA (**A**,**B**) and DA (**C**,**D**) conduits for patients 1 and 2.

**Figure 22 bioengineering-10-00135-f022:**
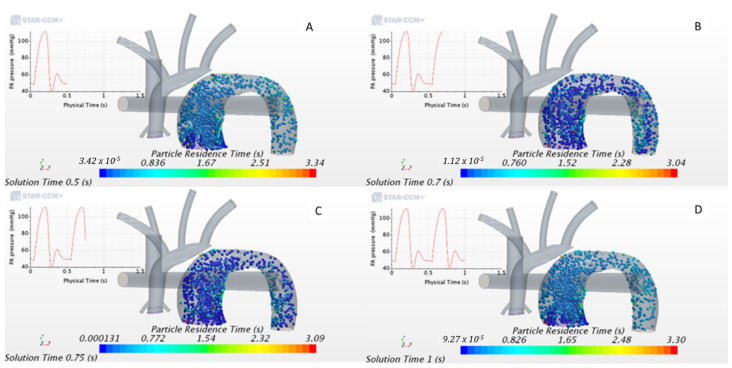
Particle residence time at different points of the cardiac cycle (**A**) = early systole, (**B**) = peak systole, (**C**) = early diastole, and (**D**) = late diastole).

**Figure 23 bioengineering-10-00135-f023:**
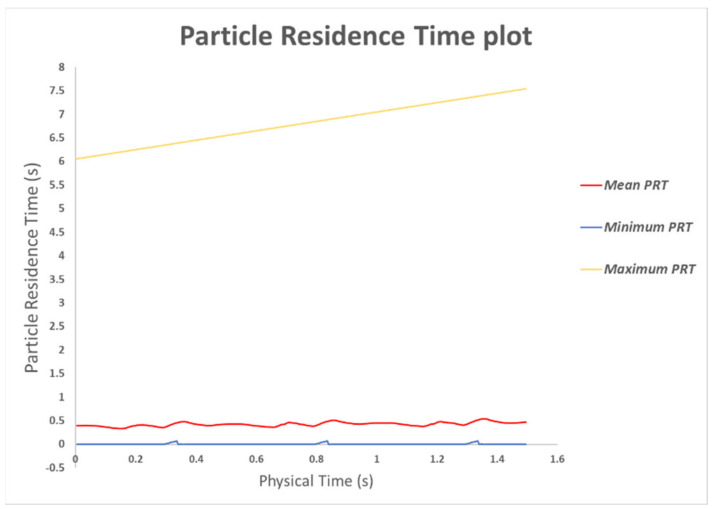
Average Particle Residence Time over three heart cycles.

**Figure 24 bioengineering-10-00135-f024:**
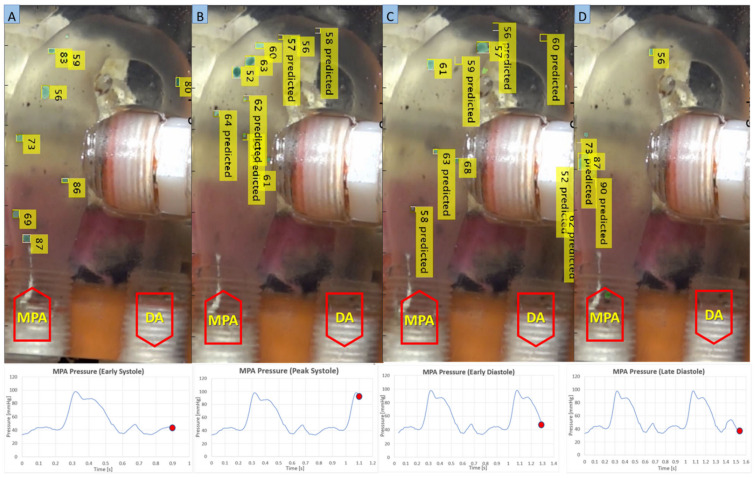
Tracked particles in the MPA conduit at different phases of the cardiac cycle (**A**) = Early systole, (**B**) = Peak systole, (**C**) = Early Diastole, (**D**) = Late Diastole).

**Figure 25 bioengineering-10-00135-f025:**
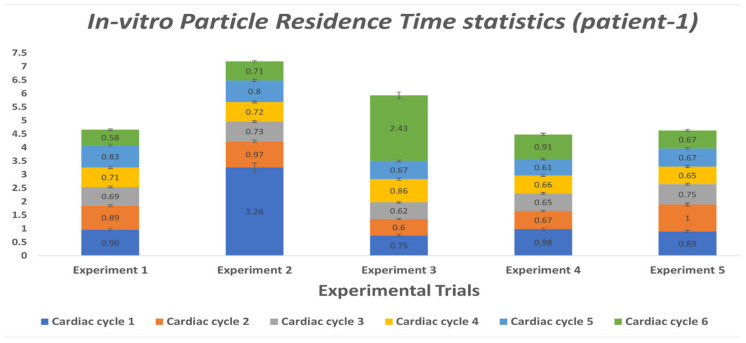
Average PRT values derived from tracked particles over six cardiac cycles in the MPA conduit for patient 1. Each stacked value represents an average PRT in each cardiac cycle.

**Figure 26 bioengineering-10-00135-f026:**
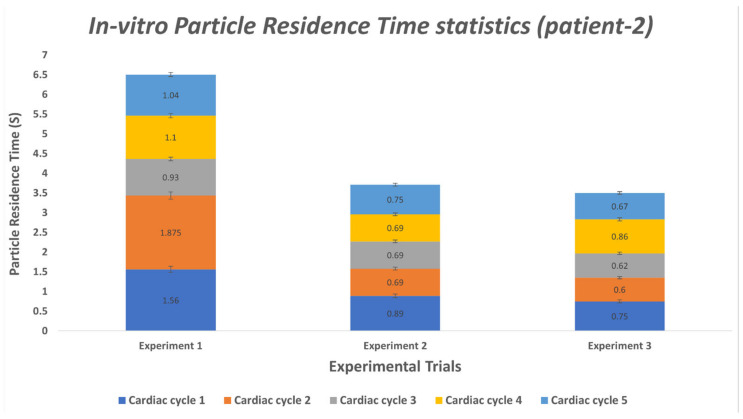
Average PRT values derived from tracked particles over six cardiac cycles in the MPA conduit for patient 1. Each stacked value represents an average PRT in each cardiac cycle.

**Figure 27 bioengineering-10-00135-f027:**
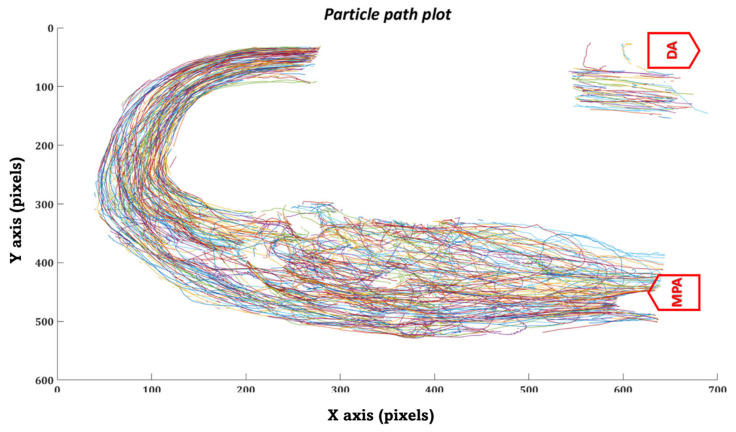
Trajectory plot of tracked particles in the MPA conduit for patient 1.

**Figure 28 bioengineering-10-00135-f028:**
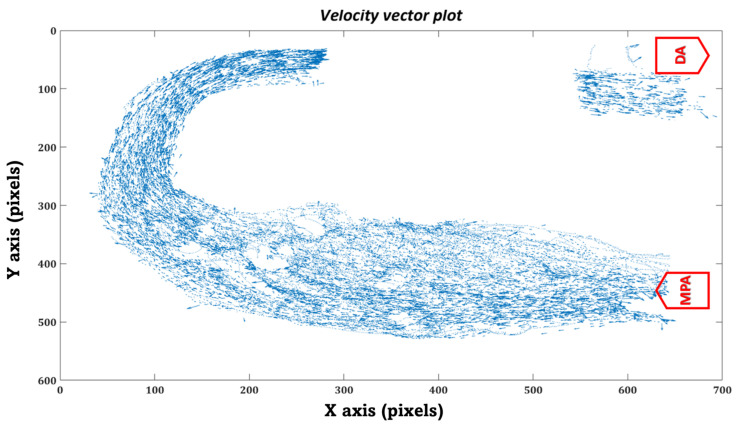
Velocity vector plot of the tracked particles in the MPA conduit for patient 1.

**Figure 29 bioengineering-10-00135-f029:**
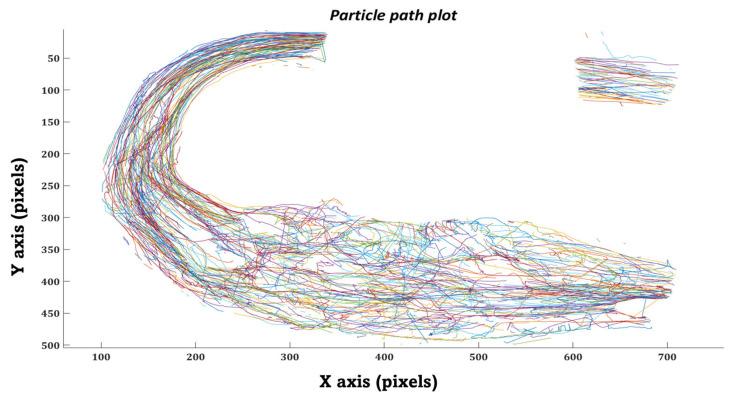
Trajectory plot of the tracked particles in the MPA conduit for patient 2.

**Figure 30 bioengineering-10-00135-f030:**
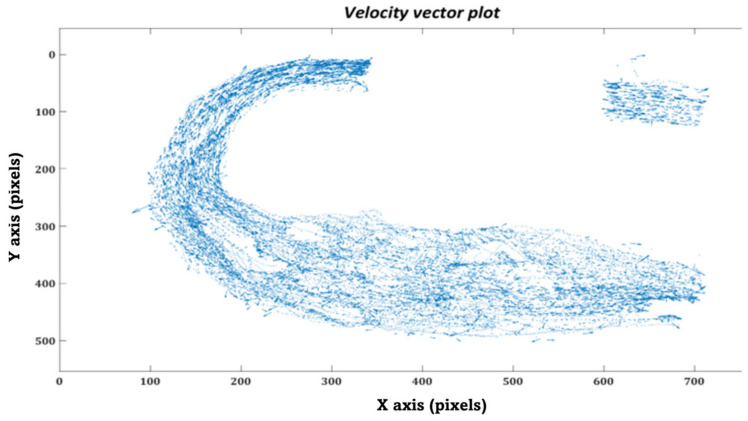
Velocity vector plot of the tracked particles in the MPA conduit for patient 2.

**Figure 31 bioengineering-10-00135-f031:**
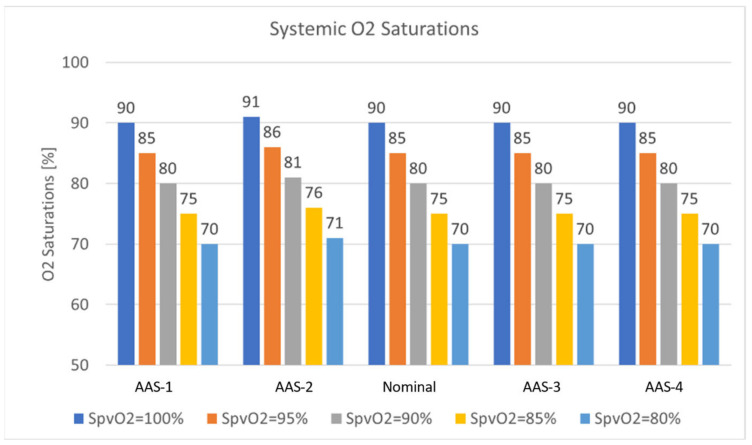
Systemic saturation with different pulmonary venous saturation as a function of aorta size.

**Figure 32 bioengineering-10-00135-f032:**
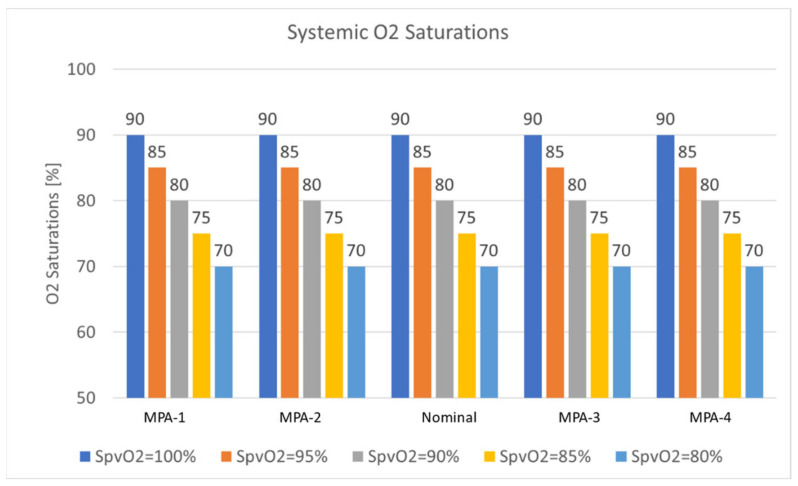
Systemic saturation with different pulmonary venous saturation as a function of the MPA narrowing.

**Figure 33 bioengineering-10-00135-f033:**
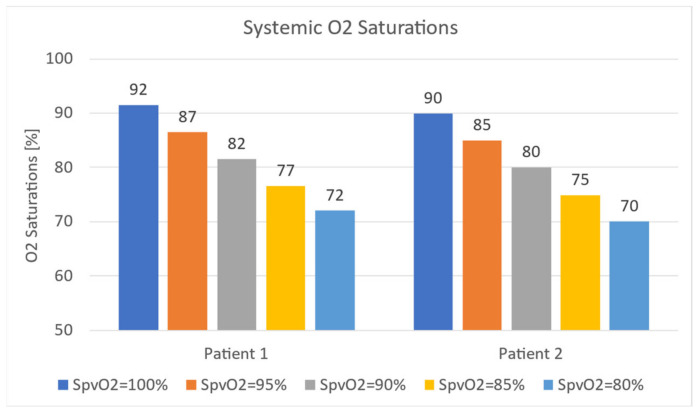
In-vitro analysis of systemic oxygen saturation for patient 1 and patient 2.

**Table 1 bioengineering-10-00135-t001:** Shows the variations of H and $D_{AA}$ for each configuration used in the in-silico study.

Case	H [mm]	Case	$D_{AA}$
MPA-1	7.23	AAS-1	5.00
MPA-2	8.19	AAS-2	6.00
Nominal	8.66	Nominal	7.00
MPA-3	9.09	AAS-3	7.50
MPA-4	9.52	AAS-4	8.50

**Table 2 bioengineering-10-00135-t002:** Describes the various types of boundary conditions imposed for each CFD region.

CFD Region	Boundary Condition
MPA	Stagnation Pressure Inlet
AA	Stagnation Pressure Inlet
DA	Mass flow outlet
SVC	Mass flow outlet
LPA	Mass flow outlet
RPA	Mass flow outlet
LcorA	Mass flow outlet
RcorA	Mass flow outlet
LSA	Mass flow outlet
RSA	Mass flow outlet
LCA	Mass flow outlet
RCA	Mass flow outlet

**Table 3 bioengineering-10-00135-t003:** Compliance values for MFL.

Compliance Sections in MFL	Compliance Value (ml/mmHg)
C_upper	0.603
C_lower	0.735
C_lpa	0.640
C_rpa	0.640

**Table 4 bioengineering-10-00135-t004:** Tabulated value for Carreau-Yasuda model constants.

Hematocrit [%]	$\mu_{\infty }$	$\mu_{o}$	λ
40	4.38989	8.4248	0.3103

**Table 5 bioengineering-10-00135-t005:** Dimensions and details of the checkerboard pattern used for calibration purposes.

Calibration Set	No. of Cells	Cell Dimension (in World Units)	Pattern Type	Mean Projection Error (in Pixel)
Calibration Set-1	5 × 6	7 mm × 7 mm	Binary	2.40
Calibration Set-2	5 × 6	7 mm × 7 mm	Color	2.26
Calibration Set-3	8 × 8	17 mm × 17 mm	Binary	10.46

**Table 6 bioengineering-10-00135-t006:** Comparison of cycle averaged flow rate [L/min] data of patient 1 between catheter reports, in-vitro and in-silico measurements.

MFL Compartment	Catheter	In-vitro	In-silico
CO [L/min]	2.99	3.15	2.99
$Q_{U}$ [L/min]	1.25	1.29	1.19
$Q_{L}$ [L/min]	1.74	1.86	1.80
$Q_{\mathrm{LPA}}$ [L/min]	0.63	0.64	0.60
$Q_{\mathrm{RPA}}$ [L/min]	0.63	0.64	0.60
$Q_{U}$/CO [L/min]	0.42	0.40	0.40

**Table 7 bioengineering-10-00135-t007:** Comparison of cycle averaged pressure [mmHg] data of patient 1 between catheter reports, in-vitro and in-silico measurements.

MFL Compartment	Catheter	In-vitro	In-silico
MPA [mmHg]	65	62	66
AA [mmHg]	65	65	65
DA [mmHg]	58	59	54
SVC [mmHg]	9	9	9
RPA [mmHg]	9	9	9
LPA [mmHg]	8	8	8

**Table 8 bioengineering-10-00135-t008:** Comparison of cycle averaged flow rate [L/min] of patient 2 between catheter reports and in-vitro measurements.

MFL Compartment	Catheter	In-vitro
CO [L/min]	1.76	1.81
$Q_{U}$ [L/min]	1.00	1.07
$Q_{L}$ [L/min]	0.76	0.74
$Q_{\mathrm{LPA}}$ [L/min]	0.50	0.52
$Q_{\mathrm{RPA}}$ [L/min]	0.50	0.52
$Q_{U}$/CO [L/min]	0.57	0.59

**Table 9 bioengineering-10-00135-t009:** Comparison of cycle averaged pressure [mmHg] of patient 2 between catheter reports and in-vitro measurements.

MFL Compartment	Catheter	In-vitro
MPA [mmHg]	52	50
AA [mmHg]	48	49
DA [mmHg]	50	49
SVC [mmHg]	15	14
RPA [mmHg]	9	9
LPA [mmHg]	8	9

**Table 10 bioengineering-10-00135-t010:** Statistical analysis using paired t-test between catheter report and in-vitro measurements on cycle averaged flow rate and pressure findings for patient-1 and patient-2.

Hemodynamic Variables	Statistical Analysis	Patient 1	Patient 2
Flow rate [L/min]	Pearson Correlation	0.976	0.916
	P(T < t) Two tail	0.854	0.170
Pressure [mmHg]	Pearson Correlation	0.998	0.998
	P(T < t) Two tail	0.675	0.713

**Table 11 bioengineering-10-00135-t011:** Statistical analysis using paired t-test between in-silico and in-vitro measurements on cycle averaged flow rate and pressure findings for patient-1 and patient-2.

Hemodynamic Variables	Statistical Analysis	Patient 1
Flow rate [L/min]	Pearson Correlation	0.999
	P(T < t) Two tail	0.657
Pressure [mmHg]	Pearson Correlation	0.995
	P(T < t) Two tail	0.872

**Table 12 bioengineering-10-00135-t012:** Experimental PRT measured in second(s) for each cardiac cycle for patient 1.

Cardiac Cycle	Experiment 1	Experiment 2	Experiment 3	Experiment 4	Experiment 5
1	0.96	3.26	0.75	0.98	0.89
2	0.89	0.97	0.60	0.67	1
3	0.69	0.73	0.62	0.65	0.75
4	0.71	0.72	0.86	0.66	0.65
5	0.83	0.80	0.67	0.61	0.67
6	0.58	0.71	2.43	0.91	0.67

**Table 13 bioengineering-10-00135-t013:** Experimental PRT measured in second (s) for each cardiac cycle for patient 2.

Cardiac Cycle	Experiment 1	Experiment 2	Experiment 3
1	1.56	0.89	0.75
2	1.875	0.69	0.60
3	0.93	0.69	0.62
4	1.10	0.69	0.86
5	1.04	0.75	0.67

**Table 14 bioengineering-10-00135-t014:** Power Loss (PL) and Efficiency in the MPA and DA conduits as a function of split ratio.

Split Ratio	MPA Power [mW]	DA Power [mW]	PL [mW]
50/50	0.338	0.307	10.33
60/40	0.414	0.376	12.66

**Table 15 bioengineering-10-00135-t015:** Power Loss (PL) and Efficiency (ƞ) as a function of H, *M*PA-PA, DA, and ƞ.

Case	H [mm]	PA [mW]	DA [mW]	PL [mW]	Ƞ
No baffle	N/A	0.416	0.38	10.33	0.97
MPA-1	7.23	0.418	0.381	12.66	0.94
MPA-2	8.19	0.417	0.379	12.66	0.94
Nominal	8.66	0.413	0.375	12.33	0.97
MPA-3	9.09	0.411	0.374	12.33	0.97
MPA-4	9.52	0.407	0.37	12.33	0.97

**Table 16 bioengineering-10-00135-t016:** Total Systemic Flow Rate [L/min] as a function of D_AA_ and flow rates at each outlet boundary CO, DA, LcorA, RcorA, LCA, RCA, RSA, and RcorA.

Case	CO [L/min]	DA [L/min]	LCA [L/min]	LSA [L/min]	LcorA [L/min]	RCA [L/min]	RSA [L/min]	RcorA [L/min]
AAS-1	2.94	1.854	0.221	0.329	0.048	0.255	0.152	0.077
AAS-2	2.97	1.826	0.131	0.253	0.049	0.383	0.251	0.073
Nominal	2.93	1.835	0.204	0.301	0.048	0.278	0.187	0.075
AAS-3	2.92	1.840	0.220	0.323	0.048	0.249	0.157	0.078
AAS-4	2.93	1.839	0.213	0.331	0.048	0.267	0.152	0.077

**Table 17 bioengineering-10-00135-t017:** Total Systemic Flow Rate [L/min] as a function of H and the flow rates at each outlet boundary: CO, DA, LcorA, RcorA, LCA, RCA, RSA, and RcorA.

Case	CO [L/min]	DA [L/min]	LCA [L/min]	LSA [L/min]	LcorA [L/min]	RCA [L/min]	RSA [L/min]	RcorA [L/min]
MPA-1	2.94	1.854	0.220	0.329	0.048	0.255	0.152	0.077
MPA-2	2.92	1.868	0.214	0.326	0.048	0.244	0.138	0.079
Nominal	2.93	1.839	0.213	0.331	0.048	0.267	0.152	0.077
MPA-3	2.91	1.824	0.218	0.328	0.048	0.256	0.155	0.078
MPA-4	2.89	1.812	0.219	0.323	0.049	0.256	0.149	0.0809

## Data Availability

Not applicable.

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
