# Peer review of "In-Silico and In-Vitro Analysis of the Novel Hybrid Comprehensive Stage II Operation for Single Ventricle Circulation"

_bioengineering, 2023, doi:10.3390/bioengineering10020135_

Round 1
Reviewer 1 Report
The reviewed paper presents up-to-date, interesting, and applicable research concerning the computational and experimental analysis of hemodynamics of the novel hybrid comprehensive stage II surgical technique. The in-silico and in-vivo techniques were utilized to investigate the effect of aortic root size, main pulmonary artery narrowing, alteration in oxygen transport, and flow distribution. The carried out research is applicable as it contributes to the improvement in surgical procedures and reduction of the mortality rate. The proposed title of the paper represents the content of the manuscript. The abstract states briefly the purpose of the research, the principal results, and the major conclusions. The authors properly presented the scientific background and the state-of-art of the investigated issues in the introduction section and referenced the up-to-date literature. The applied methodology is properly and thoroughly described. However, a more detailed description of the computational domain discretization should be provided - the authors should provide the reader with a more detailed description of the mesh distribution in the computational domain and provide more information concerning the grid-independence study. The authors are encouraged to use the Grid Convergence Index (GCI) to evaluate the preprocessing stage of the CFD analysis. The GCI is recommended by the Fluids Engineering Division of the American Society of Mechanical Engineers as a reliable method for grid independence analysis. This methodology is described in “Perspective: a method for uniform reporting of grid refinement studies,” and applied in many CFD research e.g. “Evaluation of Heat Transfer Performance of a Multi-Disc Sorption Bed Dedicated for Adsorption Cooling Technology”. The obtained results are adequately discussed and clearly presented. The conclusions are correctly drawn based on the obtained results. Moreover, the authors indicated very interesting plans for future work. The acronyms section is given in the supplementary materials but I recommend moving it to the paper itself.
Therefore, I recommend minor revision and supplementing the manuscript with a more detailed description of the computational domain discretization with grid independence analysis and moving the acronyms section to the manuscript.
Author Response
Thank you for your valuable time, please see the attachment.

Reviewer 2 Report
Overall, this paper presented a lot of useful information and should be published. I thought the analysis of the data could be improved and I make suggestions for that below.
Line 185: patent should be patient
Line 191: I'm not sure what an "eigenfeature" is.
On many of the figures (Fig. 5 for example), the labels are too small or unclear. The figure quality is poor in particular when you try to zoom in to read text.
Line 252-254: The definition of the boundary conditions is a little unclear. I assume these are the names used by star-ccm+? Can you more accurately describe what was specified at each boundary: mass flow inlet seems obvious, but mass flow outlet I assume an exit pressure was specified.
In Figure 8, no units were specified for the horizontal axis. I assume it is time in seconds?
Line 293: The boundary conditions seem to be given again, but now they just say they are of Neumann type. Please provide a better description. What is actually specified at each boundary?
Line 324: The reference to Figure 10 should be to Figure 11.
Equation 13: It is a little weird to call this energy. It is an energy flux (Watts not Joules)
Line 405: The particles were injected on the MPA inlet surface to detect recirculation zones, but this will not work. If there is a recirculation zone, particles injected from the surface will follow streamlines and thus will never enter a recirculation zone. I don't think you can use the PRT to determine whether there are recirculation zones present.
Line 409: I was confused by the word "following" in "zero injection following at random"
Line 422: I don't think the abbreviation CO was defined. (I could be wrong though).
Line 444: BMP should be BPM
Line 547: "The mean projection error for each error," Something is wrong here.
Line 489: The paper went into details about the camera set up and associated errors without really explaining what the camera was going to be used for. I think maybe talk about the particles first so the reader has some idea how this is going to be used.
Line 584 (& 729: "We have used a small roe of 2mm" What is the meaning of roe?
Line 608: The statistical analysis section needs some introduction to explain what this is going to be used for before jumping into the mathematical definitions.
Figure 25: Using an average particle residence time is not a good way to determine whether there are recirculation zones. If most of the particles pass around a recirculation eddy but a few get caught in the eddy, it will be very hard to detect that using a mean. It might be better to make a scatter plot of PRT so the scatter in the data can be assessed. This will give a sense of whether there are outliers or whether all the particles pass through in about the same time.
A more general comment is that the authors have not given any sense of the nondimensional numbers associated with the problem. In particular, I have no idea what the Reynolds number of the flow is. Examining Figures 29 & 30, I wonder if this is transition to turbulence. Is the flow chaotic or repeatable each cycle?
Similarly for PRT, it would be nice to non-dimensionalize the PRT with the mean flow time through the section. That way I could easily determine if the particles were following a nice uniform flow or if there was a lot of meandering in the particle paths.
Discussion of Tables 17 & 18 use percentage changes, but those percentages are not listed in the table so it is hard to understand what is being discussed. It would be better to refer directly to the numbers in the table and then give the percentage changes.
Author Response

(The authors gave the same response as above.)

Reviewer 3 Report
This paper presents a computational and benchtop study of a new surgical procedure for treating single ventricle congenital heart defects. The paper builds upon previous work in this area by the same group.
The paper is well written with good coverage of the existing literature and sufficient methodological detail. At a technical level the work seems to be correct, though there are a few questions. Readability would improve from some clarifications and a reduction in length. Detailed comments are below.
1) A strength of the paper is the inclusion of both simulation and experimental data. However there does not seem to be an attempt at direct comparison between the two. This is surprising since the computed and measured values for the particle residence time and fluid pressures are available. Can the authors comment on why such comparisons are not provided?
2) Sec 3.4.2 presents oxygen saturation values in from the in vitro (experimental) study. How was oxygen consumption implemented in the experiments? What is the meaning of saturation here since the fluid is a water-glycerin mixture that does not contain hemoglobin?
3) Lines 753 – 758 assert that the existence of two flow regimes in different spatial locations. This is not obvious from the associated images (Figures 27-30). What is the quantity that is plotted in Figures 28 and 30? It is difficult to discern any patterns from the particle tracks shown in Figures 27 and 29. Further, the orientation of these figures seems to be rotated by 90 degrees from Figures 24 and 26 (but in the same orientation as Figure 21). It would be helpful to label landmarks for clarity.
4) Are Figure 27-30 from the experiments? The presence of a blank region between at the top half suggests that this is the case, but it is not clear from the text whether these are experiments of simulations.
5) The coupling between the lumped parameter and CFD models should be explained better.
a) The governing equations for the lumped parameter model involve the equivalent of Kirchoff’s current and voltage laws. Some description of the form of these equations should be given.
b) It is said that the right ventricle (RV – this acronym should be expanded in line 234) powers the circuit using a time varying elastance. An input pressure is also needed. What form does this take?
c) It is stated that the BCs are of Neumann type (line 293). Aren’t the pressures and flowrates determined by the lumped parameter model specified at the CFD domain boundaries? These would be Dirichlet conditions. What conditions are imposed at the venous end of the lumped parameter model?
d) Line 296 states the CFD outputs are used to compute flow resistances as inputs to the lumped parameter model. It is not clear how this works. If the resistances are not constant their flow dependence should be specified. Do the flow and pressure from the CFD not link to the lumped parameter model? Why?
e ) The arrows in Fig 10 seem to point the wrong way: the output of the LPM are BCs that feed into the CFD model.
f) A higher quality image is needed for Fig 7 – elements of the model are not clear
6) An equation for the Carreau-Yasuda model should be given and the units specified in Table 3. The table caption should be corrected.
7) The section on the calibration and error estimation of the imaging setup could be moved to an Appendix to reduce the length of the paper. Similarly Tables 13 and 14 can be omitted.
8) Has any validation of the Kalman filtering technique for particle been carried out?
9) What is the density of the glycerin – water mixture and the density of the roe used in the in vitro studies? It is important to ensure neutral buoyance to replicate the massless particles of the CFD simulations.
10) Figure 24 is confusing. Particle residence time is the time duration from the instant the particle enters the domain to the instant it exits. It does not make sense to define a residence time while the particle is still traversing the domain.
11) Is Fig 26 showing sequential positions of tracked particles? The labels are not clear.
12) For Figure 1 expand the acronyms in the figure caption. This will make it easier for a reader to understand the figure without having to search the text for the expansions. In Fig 3 label DAA. Table 12 is probably for patient 2 and not patient 1.

Author Response

(The authors gave the same response as above.)

Round 2
Reviewer 3 Report
The authors have addressed the issues raised in the review. I recommend that the paper be accepted.